# Homomorphic Matrix Completion

**Xiao-Yang Liu**[1*], **Zechu Li**[2*] **Xiaodong Wang**[1]
[1]Department of Electrical Engineering, Columbia University, New York,
[2]Department of Computer Science, Columbia University, New York,
{xl2427, zl2993, xw2008}@columbia.edu

## Abstract

In recommendation systems, global positioning, system identification, and mobile social networks, it is a fundamental routine that a server completes a low-rank matrix from an observed subset of its entries. However, sending data to a cloud server raises up the data privacy concern due to eavesdropping attacks and the single-point failure problem, e.g., the Netflix prize contest was canceled after a privacy lawsuit. In this paper, we propose a homomorphic matrix completion algorithm for privacy-preserving purpose. First, we formulate a *homomorphic matrix completion* problem where a server performs matrix completion on cyphertexts, and propose an encryption scheme that is fast and easy to implement. Secondly, we prove that the proposed scheme satisfies the *homomorphism property* that decrypting the recovered matrix on cyphertexts will obtain the target matrix (on plaintexts). Thirdly, we prove that the proposed scheme satisfies an $(\epsilon, \delta)$-differential privacy property. While with similar level of privacy guarantee, we reduce the best-known error bound $O(\sqrt[10]{n_1^3 n_2})$ to EXACT recovery at a price of more samples. Finally, on synthetic data and real-world data, we show that both homomorphic nuclear-norm minimization and alternating minimization algorithms achieve accurate recoveries on cyphertexts, verifying the homomorphism property.

## 1 Introduction

The recurring low-rank matrix completion problem [4, 6, 22, 30, 13, 29, 46] concerns completing a low-rank matrix from a randomly observed subset of entries. It has wide applications in recommendation systems (collaborative filtering) [1, 47, 26], computer vision [2, 16, 27], global positioning [48, 32], sensory data analysis in Internet of Things [25, 34, 33], system identification, network data analysis [51, 11], mobile social networks [23, 38], etc. Existing works [6, 9] have demonstrated a remarkable fact: if an $n \times n$ matrix with rank $r \ll n$ satisfies a certain incoherence property, then with high probability, it is possible to exactly recover the matrix from $O(nr \, \mathbf{poly} \log n) \ll n^2$ entries using polynomial-time algorithms. Intuitively, one needs roughly $(2nr - r^2)$ parameters [6] (this is the dimension of the tangent space to the manifold of rank-$r$ matrices) to fix an $n \times n$ matrix of rank $r$, and the sampling randomness introduces a $\log n$ factor due to a coupon collector's effect. The information theoretical lower bound is $\Omega(nr \log n)$ [6], while the tightest known upper bound is $O(nr \log^2 n)$ [9] with another $\log n$ factor from the Golfing scheme used by the recovery algorithms.

The low-rank matrix completion problem usually deals with large-scale matrices that involve extensive computations, while in mobile computing, smart devices usually outsource such a huge computation task to a cloud server. However, sending data to a server or publishing anonymized data raises up privacy concerns [23, 44, 42], e.g., the recommendation contest Netflix prize was canceled after privacy lawsuit [35]. There are two major obstructive factors: anonymization in data publishing is still vulnerable, and storing sensitive data on a cloud server may encounter the single-point of failure

---

[*]Equal contribution.

36th Conference on Neural Information Processing Systems (NeurIPS 2022).

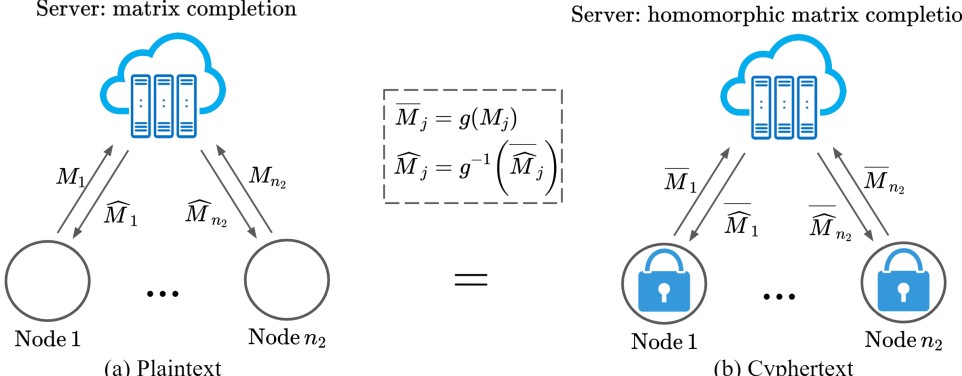

Figure 1: Matrix completion on plaintext VS. homomorphic matrix completion on cyphertext.

(SPOF) problem, say hackers. Existing works [20, 19, 10] address the privacy concern in various ways, e.g., a popular approach is to add noise to the data [20], therefore making a tradeoff between the recovery accuracy and the level of privacy.

In cloud computing and distributed systems, the homomorphism property [14, 45] allows computations to be carried out on cyphertexts, generating an encrypted result which, when decrypted, matches the result of operations performed on the corresponding plaintexts. In this manner, **homomorphic encryption securely chains together different services without sacrificing recovery accuracy, but may at a price of extensive computation**. There are several partially homomorphic crypto-systems, and also a number of fully homomorphic crypto-systems [14, 45]. In addition, the homomorphic property can also be used to create many other secure systems, for example secure voting systems, collision-resistant hash functions, private information retrieval schemes [43], etc.

In this paper, we integrate the large-scale distributed matrix completion task with a homomorphic encryption-decryption scheme, which guarantees the EXACT recovery and differential privacy at a price of more samples. First, we define the *homomorphic matrix completion problem* that ensures data privacy by preserving a homomorphism property between plaintexts and cyphertexts. Specifically, we propose a homomorphic encryption-decryption scheme, in which each node performs local encryption and decryption, and uploads an encrypted incomplete vector to a server that carries out the matrix completion computation. Then, we theoretically prove that the proposed scheme satisfies the homomorphism and differential privacy properties — reducing the best-known error bound $O(\sqrt[10]{n_1^3 n_2})$ [20] to EXACT recovery. Finally, based on synthetic and real-world data, we show that the homomorphic nuclear-norm minimization and alternating minimization algorithms achieve accurate recoveries on both cyphertexts and plaintexts, verifying the homomorphism property.

## 2 Homomorphic Matrix Completion Problem

First, we formally define the homomorphic matrix completion problem. Then, we introduce a notation of privacy by adapting the join $(\epsilon, \delta)$ differential privacy, which is a subspace-aware variant.

### 2.1 Notations and Preliminaries

**Notations**: Let $e_i$ denote the $i$-th standard basis, $I_k$ denote the $k \times k$ identity matrix, and $\mathcal{I}$ denote the identity linear operator. For matrix $X$, the $(i, j)$-th element is $X_{ij}$ or $X(i, j)$, the $j$-th column is $X_j$, and the transpose is $X^\top$. The concatenation of two matrices $A \in \mathbb{R}^{n_1 \times n_2}$ and $B \in \mathbb{R}^{n_1 \times n_3}$ with the same number of rows is denoted by $[A, B] \in \mathbb{R}^{n_1 \times (n_2 + n_3)}$. By *with high probability* (w.h.p.) we mean that with probability at least $1 - c_1 n^{-c_2}$ for some positive constants $c_1, c_2$. Let $\mathcal{N}(0, \sigma^2)$ denote a Gaussian distribution with mean 0 and standard deviation $\sigma$. We use an overline to represent the encrypted version of a variable: variables before encryption are called *plaintexts*, e.g., $X$, while the encrypted variables are called *cyphertexts*, e.g., $\overline{X}$.

Let $M_\Omega \in \mathbb{R}^{n_1 \times n_2}$ denote the observed entries of a data matrix $M \in \mathbb{R}^{n_1 \times n_2}$, where $\Omega \subseteq \{(1, 1), (1, 2), ..., (n_1, n_2)\}$ indicates the observed entries. We define a linear operator

$\mathcal{P}_\Omega : \mathbb{R}^{n_1 \times n_2} \to \mathbb{R}^{n_1 \times n_2}$ to represent the partial observation model as follows

$$[\mathcal{P}_\Omega(\boldsymbol{M})]_{ij} = \begin{cases} \boldsymbol{M}_{ij}, \text{ if } (i,j) \in \Omega \\ \quad 0, \text{ otherwise.} \end{cases} \tag{1}$$

Assuming the true data matrix $\boldsymbol{M}$ is low-rank, i.e., $\text{rank}(\boldsymbol{M}) = r \ll \min(n_1, n_2)$. The (compact) singular value decomposition (SVD) is $\boldsymbol{M} = \boldsymbol{U}\boldsymbol{\Sigma}\boldsymbol{V}^\top$, where $\boldsymbol{U} \in \mathbb{R}^{n_1 \times r}$ represents $r$ left singular vectors (a basis of the column subspace), $\boldsymbol{V} \in \mathbb{R}^{n_2 \times r}$ represents $r$ right singular vectors (a basis of the row subspace), and $\boldsymbol{\Sigma} = \text{diag}([\sigma_1, \sigma_1, \cdots, \sigma_r]) \in \mathbb{R}^{r \times r}$ with singular values $\sigma_1 \geq \sigma_2 \geq \cdots \geq \sigma_r \geq 0$. The $\ell_2$-norm of a vector is $||\boldsymbol{x}||_2$, while the Frobenius norm and nuclear norm of $\boldsymbol{M}$ are $||\boldsymbol{M}||_F = \sqrt{\sum_{i,j} |\boldsymbol{M}_{ij}|^2}$ and $||\boldsymbol{M}||_* = \sum_{i=1}^r \sigma_i$, respectively. The operator norm (spectral norm) of a matrix and a linear operator $\mathcal{L}$ are defined as follows

$$||\boldsymbol{M}|| \triangleq \sup_{\boldsymbol{x} \in \mathbb{R}^{n_2}, \, ||\boldsymbol{x}||_2 \leq 1} ||\boldsymbol{M}\boldsymbol{x}||_2 = \sigma_1, \text{ and } ||\mathcal{L}|| \triangleq \sup_{||\boldsymbol{X}||_F \leq 1} ||\mathcal{L}(\boldsymbol{X})||_F. \tag{2}$$

**Definition 1.** *(Column subspace and null space [28]) Let $\boldsymbol{A} \in \mathbb{R}^{n_1 \times n_2}$. The set $\mathcal{S}(\boldsymbol{A}) = \{\boldsymbol{b} \in \mathbb{R}^{n_1} \,|\, \boldsymbol{b} = \boldsymbol{A}\boldsymbol{x}, \ \boldsymbol{x} \in \mathbb{R}^{n_2}\}$ is the column space or range of $\boldsymbol{A}$, and the set $\boldsymbol{Ker}(\boldsymbol{A}) = \{\boldsymbol{x} \in \mathbb{R}^{n_2} \,|\, \boldsymbol{A}\boldsymbol{x} = \boldsymbol{0}\}$ is the kernel or (right) null space of $\boldsymbol{A}$.*

The null space (kernel space) of operator $\mathcal{P}_\Omega$ is $\boldsymbol{Ker}(\mathcal{P}_\Omega) = \{\boldsymbol{Z} \in \mathbb{R}^{n_1 \times n_2} \,|\, \mathcal{P}_\Omega(\boldsymbol{Z}) = \boldsymbol{0}\}$, which is denoted as $\Omega^\perp$ [2]. Let $\Omega \sim \boldsymbol{Uni}(m)$ denote a set with $m$ entries, which is sampled uniformly from all sets of $m$ entries, and $\Omega \sim \boldsymbol{Ber}(p)$ denote a set with $\mathbb{E}|\Omega| = m$ entries, where each entry is sampled independently according to a Bernoulli model with $p = m/(n_1 n_2)$.

Let $\boldsymbol{P_U}$ and $\boldsymbol{P_V}$ denote the orthogonal projections onto the column and row space of $\boldsymbol{M}$, respectively,

$$\boldsymbol{P_U} = \sum_{i \in [r]} \boldsymbol{u}_i \boldsymbol{u}_i^\top = \boldsymbol{U}\boldsymbol{U}^\top, \quad \boldsymbol{P_V} = \sum_{i \in [r]} \boldsymbol{v}_i \boldsymbol{v}_i^\top = \boldsymbol{V}\boldsymbol{V}^\top. \tag{3}$$

Define an orthogonal decomposition $\mathbb{R}^{n_1 \times n_2} = T \oplus T^\perp$, where $T$ is the linear space spanned by matrices with the same column space or row space as $\boldsymbol{M}$, and $T^\perp$ is its orthogonal complement that consists of matrices with row-space orthogonal to the row-space $\boldsymbol{V}$ and column-space orthogonal to the column-space $\boldsymbol{U}$. $T$ can be expressed as follows

$$T = \{\boldsymbol{U}\boldsymbol{A}^\top + \boldsymbol{B}\boldsymbol{V}^\top \,|\, \boldsymbol{A} \in \mathbb{R}^{n_1 \times r}, \ \boldsymbol{B} \in \mathbb{R}^{n_2 \times r}\}. \tag{4}$$

The orthogonal projection $\mathcal{P}_T$ onto $T$ and the orthogonal projection onto $T^\perp$ are as follows

$$\begin{aligned} \mathcal{P}_T(\boldsymbol{X}) &= \boldsymbol{P_U}\boldsymbol{X} + \boldsymbol{X}\boldsymbol{P_V} - \boldsymbol{P_U}\boldsymbol{X}\boldsymbol{P_V}, \\ \mathcal{P}_{T^\perp}(\boldsymbol{X}) &= (\mathcal{I} - \mathcal{P}_T)(\boldsymbol{X}) = (\boldsymbol{I}_{n_1} - \boldsymbol{P_U})\boldsymbol{X}(\boldsymbol{I}_{n_2} - \boldsymbol{P_V}). \end{aligned} \tag{5}$$

## 2.2 Problem Formulation for Homomorphic Matrix Completion

We are interested in completing large-scale matrices and want to outsource this compute-intensive task from mobile devices to a cloud server. Here we aim to preserve the matrix entries from leakage, which is the key concern for recommendation systems as in Netflix's privacy lawsuit [35].

**Distributed matrix completion problem on plaintexts**. Assume that there are $n_2$ nodes with limited computing power and a cloud server with superior computing power. The $j$-th node has an attribute vector $\boldsymbol{M}_j \in \mathbb{R}^{n_1}$, $j = 1, ..., n_2$, however, it is incomplete and the observed entries are indexed by a set $\Omega_j \subseteq \{(1, j), (2, j), ..., (n_1, j)\}$. We assume that the true values of these $n_2$ vectors form a low-rank matrix $\boldsymbol{M} \in \mathbb{R}^{n_1 \times n_2}$ with rank $r \ll \min(n_1, n_2)$, the $\ell_2$-norms of the attribute vectors is bounded by $L$, i.e., $\max_{1 \leq j \leq n_2} ||\boldsymbol{M}_j||_2 \leq L$, and the observation set $\Omega = \bigcup_{j=1,...,n_2} \Omega_j$. Nodes upload their incomplete vectors to a cloud server to carry out a matrix completion task

$$\text{Find a matrix } \boldsymbol{X} \in \mathbb{R}^{n_1 \times n_2}, \text{ s.t. } \mathcal{P}_\Omega(\boldsymbol{X}) = \mathcal{P}_\Omega(\boldsymbol{M}), \text{ rank}(\boldsymbol{X}) \leq r, \tag{6}$$

where $\Omega \sim \boldsymbol{Uni}(m)$ and $r$ may be unknown. Without loss of generality, we assume that $n_1 \leq n_2$ from now on. Note that our formulation also includes the case [37] where a matrix is distributed into blocks and then is completed in parallel.

---

[2] We adopt the notation $\Omega^\perp$ since $\boldsymbol{Ker}(\mathcal{P}_\Omega)$ corresponds to a set of matrices vanishing at $\Omega^\perp$.

**Homomorphic matrix completion problem on cyphertexts**. In cloud computing, the homomorphism property allows computations to be carried out on cyphertexts, generating an encrypted result which, when decrypted, matches the result of operations performed on the plaintext. Following this paradigm, we define a homomorphic matrix completion problem that ensures data privacy. As shown in Fig. 1, this novel framework consists of three main steps:

1) each node locally encrypts as $\mathcal{P}_{\Omega_j}(\overline{M}_j) = \mathcal{P}_{\Omega_j}(g(M_j))$ with its private keys, $j = 1, ..., n_2$, and uploads $\mathcal{P}_{\Omega_j}(\overline{M}_j)$ to a cloud server that later forms an incomplete matrix $\mathcal{P}_{\Omega}(\overline{M}) \in \mathbb{R}^{n_1 \times n_2}$;

2) the cloud server solves the following matrix completion problem given $\mathcal{P}_{\Omega}(\overline{M})$ and sends back the recovered vector $\widehat{\overline{M}}_j$ to the $j$-th node, $j = 1, ..., n_2$,

$$\text{Find a matrix } \overline{X} \in \mathbb{R}^{n_1 \times n_2}, \ \text{ s.t. } \mathcal{P}_{\Omega}(\overline{X}) = \mathcal{P}_{\Omega}(\overline{M}), \ \text{rank}(\overline{X}) \leq \overline{r}, \tag{7}$$

where $\overline{r} = \text{rank}(\overline{M})$ may be slightly bigger than $r$ due to by the encryption scheme $g(\cdot)$.

3) each node locally decrypts its own vector using private keys, i.e., $\widehat{M}_j = g^{-1}(\widehat{\overline{M}}_j), j = 1, ..., n_2$.

## 2.3 Notions of Privacy

We introduce a new variant of differential privacy for low-rank matrices.

### 2.3.1 Differential Privacy

Let $D = \{d_1, ..., d_n\}$ be a dataset of $n$ entries and $\mathcal{T}$ be a fixed domain, where each entry $d_j \in \mathcal{T}$ encodes potentially sensitive information about node $j$. Let $\mathcal{A} : \mathcal{T}^n \to \mathcal{O}^n$ be an algorithm that operates on dataset $D$ and produces $n$ outputs, one for each node $j$ and from a set of possible output $\mathcal{O}$. Let $D_{-j}$ denote the dataset $D$ without the entry of the $j$-th node, and similarly $\mathcal{A}_{-j}(D)$ denote the set of outputs without the output for the $j$-th node. Let $(d_j; D_{-j})$ denote the dataset obtained by adding a data entry $d_j$ to the dataset $D_{-j}$.

The $(\epsilon, \delta)$-differential privacy and joint $(\epsilon, \delta)$-differential privacy [21] are given in the following.

**Definition 2.** *($(\epsilon, \delta)$-differential privacy [12]). An algorithm $\mathcal{A}$ satisfies $(\epsilon, \delta)$-differential privacy if for any node $j$, any two possible values of data entry $d_j, d'_j \in \mathcal{T}$ for node $j$, any tuple of data entries for all other nodes $D_{-j} \in \mathcal{T}^{n-1}$, and any output set $O \subseteq \mathcal{O}^n$, we have*

$$\mathbb{P}_{\mathcal{A}}[\mathcal{A}(d_j; D_{-j}) \in O] \leq e^{\epsilon} \cdot \mathbb{P}_{\mathcal{A}}[\mathcal{A}(d'_j; D_{-j}) \in O] + \delta. \tag{8}$$

**Definition 3.** *(Joint $(\epsilon, \delta)$-differential privacy [21]). An algorithm $\mathcal{A}$ satisfies $(\epsilon, \delta)$-joint differential privacy if for any node $j$, any two possible values of data entry $d_j, d'_j \in \mathcal{T}$ for node $j$, any tuple of data entries for all other nodes $D_{-j} \in \mathcal{T}^{n-1}$, and any output set $O \subseteq \mathcal{O}^{n-1}$, we have*

$$\mathbb{P}_{\mathcal{A}}[\mathcal{A}_{-j}(d_j; D_{-j}) \in O] \leq e^{\epsilon} \cdot \mathbb{P}_{\mathcal{A}}[\mathcal{A}_{-j}(d'_j; D_{-j}) \in O] + \delta. \tag{9}$$

Intuitively, an algorithm $\mathcal{A}$ satisfies $(\epsilon, \delta)$-differential privacy if for any node $j$ and dataset $D$, $\mathcal{A}(D)$ and $D_{-j}$ do not reveal "much" information about $d_j$. For low-rank matrices, [20] used a relaxed notion *joint $(\epsilon, \delta)$-differential privacy*: an algorithm $\mathcal{A}$ satisfies joint $(\epsilon, \delta)$-differential privacy if for any node $j$ and dataset $D$, $\mathcal{A}_{-j}(D)$ (the output for the other $n - 1$ nodes) and $D_{-j}$ (data entries of the other $n - 1$ nodes) do not reveal "much" information about $d_j$. Relaxing $(\epsilon, \delta)$-differential privacy to joint $(\epsilon, \delta)$-differential privacy is reasonable for the matrix completion problem since the $j$-th column for the $j$-th node can reveal a lot of information about $d_j$.

### 2.3.2 Differential Privacy for Low-rank Matrix Completion

We would like to point out that joint $(\epsilon, \delta)$-differential privacy in Def. 3 can be further refined. For a low-rank matrix $M$, its column subspace $\mathcal{S}(M)$ is *global information*, which is shared across all $n_2$ nodes and can be easily inferred from $\mathcal{A}_{-j}(D)$ and $D_{-j}$. Note that the differential privacy notion aims to protect individual information, rather than global information. We adapt it to low-rank matrices by excluding the shared column subspace.

Low-rank matrices have linearly dependent columns, and this dependency is reflected in the fact that they share a common column subspace. Formally, a rank-$r$ matrix $M = U\Sigma V^{\top}$ can be expressed

---

**Algorithm 1** Homomorphic matrix completion at the cloud server

---

**Input**: parameters $n_1, n_2, r, k$.

**Output**: matrix $\boldsymbol{K} \in \mathbb{R}^{n_1 \times k}$ as public keys, the recovered matrix $\overline{\widehat{\boldsymbol{X}}} \in \mathbb{R}^{n_1 \times n_2}$ (cyphertexts).

1: Generate a random matrix $\boldsymbol{K} \in \mathbb{R}^{n_1 \times k}$ and broadcast $\boldsymbol{K}$ to all $n_2$ nodes;

2: **until** received all $n_2$ encrypted vectors $\mathcal{P}_{\Omega_j}(\overline{\boldsymbol{M}}_j)$ (line 4 in Alg. 2) **do**

3:   Carry out a matrix completion task in (7) and obtain $\overline{\widehat{\boldsymbol{X}}} \in \mathbb{R}^{n_1 \times n_2}$;

4:   Send the recovered vector $\overline{\widehat{\boldsymbol{X}}}_j \in \mathbb{R}^{n_1}$ back to the $j$-th node, $j = 1, ..., n_2$.

5: **end**

---

---

**Algorithm 2** Homomorphic matrix completion at node $j$, for $j = 1, ..., n_2$

---

**Input**: an incomplete vector $\mathcal{P}_{\Omega_j}(\boldsymbol{M}_j)$, observation set $\Omega_j$, and parameters $n_1, r, k$.

**Output**: an recovered vector $\widehat{\boldsymbol{X}}_j$ (plaintexts).

1: **until** received $\boldsymbol{K} \in \mathbb{R}^{n_1 \times k}$ from the server (line 1 in Alg. 1) **do**

2:   Generate $k$ random numbers $\boldsymbol{R}_j \overset{\text{i.i.d}}{\sim} \mathcal{N}(\boldsymbol{0}, \sigma^2 \boldsymbol{I}_k)$;

3:   Perform local encryption as $\mathcal{P}_{\Omega_j}(\overline{\boldsymbol{M}}_j) = \mathcal{P}_{\Omega_j}(\boldsymbol{M}_j) + \mathcal{P}_{\Omega_j}(\boldsymbol{K}\boldsymbol{R}_j)$;

4:   Upload $\mathcal{P}_{\Omega_j}(\overline{\boldsymbol{M}}_j)$ to the cloud server;

5: **end**

6: **until** received the recovered vector $\overline{\widehat{\boldsymbol{X}}}_j$ from the cloud server (line 4 in Alg. 1) **do**

7:   Using $\boldsymbol{R}_j$ and $\boldsymbol{K}$, decrypt $\overline{\widehat{\boldsymbol{X}}}_j$ to obtain $\widehat{\boldsymbol{X}}_j$, i.e., $\widehat{\boldsymbol{X}}_j = \overline{\widehat{\boldsymbol{X}}}_j - \boldsymbol{K}\boldsymbol{R}_j$.

8: **end**

---

as $\boldsymbol{M} = \boldsymbol{U}\boldsymbol{C}$ where $\boldsymbol{U} \in \mathbb{R}^{n_1 \times r}$ and $\boldsymbol{C} = \boldsymbol{\Sigma}\boldsymbol{V}^\top \in \mathbb{R}^{r \times n_2}$; alternatively, a column can be expressed as $\boldsymbol{M}_j = \boldsymbol{U}\boldsymbol{C}_j$, for $j = 1, ..., n_2$, where $\boldsymbol{C}_j$ is the coefficient vector (individual information) of the $j$-th node in the column subspace with basis $\boldsymbol{U}$ (global information).

The following subspace-aware joint $(\epsilon, \delta)$-differential privacy considers the coefficient vectors $\boldsymbol{C}_j$ for $j = 1, ..., n_2$, i.e., $D$ in Def. 3 corresponds to the coefficient matrix $\boldsymbol{C} \in \mathbb{R}^{r \times n_2}$.

**Definition 4.** *(Subspace-aware joint $(\epsilon, \delta)$-differential privacy). Assume $n_2$ nodes' data vector form a rank-$r$ matrix $\boldsymbol{M} \in \mathbb{R}^{n_1 \times n_2}$ with $\boldsymbol{M} = \boldsymbol{U}\boldsymbol{S}\boldsymbol{V}^\top = \boldsymbol{U}\boldsymbol{C}$ where $\boldsymbol{U} \in \mathbb{R}^{n_1 \times r}$ and $\boldsymbol{C} = \boldsymbol{S}\boldsymbol{V}^\top \in \mathbb{R}^{r \times n_2}$. A matrix completion algorithm $\mathcal{A}$ satisfies subspace-aware $(\epsilon, \delta)$-joint differential privacy if for any node $j$, any two possible coefficient vectors $\boldsymbol{C}_j, \boldsymbol{C}'_j \in \mathbb{R}^r$ for node $j$, any tuple of coefficient vectors for all other nodes $\boldsymbol{C}_{-j} \in \mathbb{R}^{r \times (n_2 - 1)}$, and any output set $O \subseteq \mathbb{R}^{r \times n_2}$ that consists of estimated coefficient vectors in a column subspace with basis $\boldsymbol{U}$, we have*

$$\mathbb{P}_{\mathcal{A}}[\mathcal{A}_{-j}(\boldsymbol{C}_j; \boldsymbol{C}_{-j}|\boldsymbol{U}) \in O] \leq e^\epsilon \cdot \mathbb{P}_{\mathcal{A}}[\mathcal{A}_{-j}(\boldsymbol{C}'_j; \boldsymbol{C}_{-j}|\boldsymbol{U}) \in O] + \delta. \tag{10}$$

## 3 Novel Homomorphic Framework for Matrix Completion

We propose a homomorphic encryption-decryption scheme: a node performs local encryption and decryption, and uploads an encrypted vector to a server to perform the matrix completion computation.

### 3.1 Our Idea: Hiding a Low-rank Data Matrix in a Larger Subspace

To preserve the privacy of a low-rank data matrix $\boldsymbol{M} \in \mathbb{R}^{n_1 \times n_2}$ with rank $r$, our idea is to hide $\boldsymbol{M}$ (lies in an $r$-dimensional subspace) into a larger subspace of dimension $\overline{r}$, such that $\overline{r} \geq r$ and $r, \overline{r} \ll n_1$. A sound approach would be enlarging the original subspace of the data matrix (i.e., the plaintext) as follows: a cloud server generates a random matrix $\boldsymbol{K} \in \mathbb{R}^{n_1 \times k}$ as public keys, $k \ll n_1$, and broadcasts $\boldsymbol{K}$ to all $n_2$ nodes; then, node $j$ generates $k$ random numbers as private keys $\boldsymbol{R}_j \in \mathbb{R}^k$, and encrypts its vector $\boldsymbol{M}_j \in \mathbb{R}^{n_1}$ as follows (a version with missing entries will be given in (12))

$$\overline{\boldsymbol{M}}_j = \boldsymbol{M}_j + \boldsymbol{K}\boldsymbol{R}_j, \ j = 1, ..., n_2; \quad \text{Equivalently, } \overline{\boldsymbol{M}} = \boldsymbol{M} + \boldsymbol{R}\boldsymbol{R}. \tag{11}$$

In the encryption scheme (11), $\boldsymbol{M}$ is added up with $\boldsymbol{K}\boldsymbol{R}$, resulting in a matrix $\overline{\boldsymbol{M}}$ with rank $\overline{r} \leq r + k$. Since $\overline{r} \ll n_1$, $\overline{\boldsymbol{M}}$ is also low-rank, it is possible to recover $\overline{\boldsymbol{M}}$ from a subset of entries.

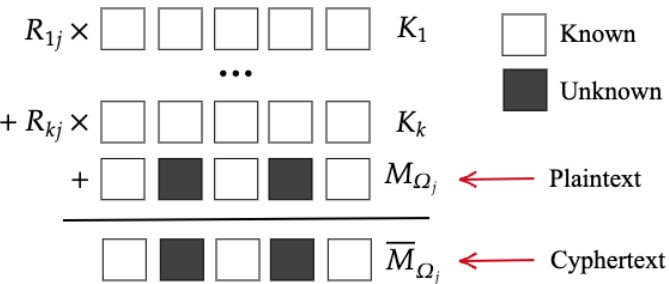

Figure 2: Our encryption method. Plaintext and cyphertext have the same set $\Omega$ of missing entries.

## 3.2 Proposed Homomorphic Encryption-Decryption Scheme

We propose a homomorphic encryption-decryption scheme that consists of the following steps, while the pseudocodes are summarzied in Alg. 1 and Alg. 2.

- First, in line 1 of Alg. 1, the cloud server generates a random matrix $\boldsymbol{K} \in \mathbb{R}^{n_1 \times k}$ as public keys, then broadcasts $\boldsymbol{K}$ to all $n_2$ nodes.

- Second, in lines 1-5 of Alg. 2, after receiving $\boldsymbol{K} \in \mathbb{R}^{n_1 \times k}$ from the server (line 1 in Alg. 1), the $j$-th node locally carries out an encryption with $k$ private keys (i.e., $\boldsymbol{R}_j \in \mathbb{R}^k$). As shown in Fig. 2, the $j$-th node locally encrypts its incomplete vector $\mathcal{P}_{\Omega_j}(\boldsymbol{M}_j)$ as follows

$$\mathcal{P}_{\Omega_j}(\overline{\boldsymbol{M}}_j) = \mathcal{P}_{\Omega_j}(\boldsymbol{M}_j) + \mathcal{P}_{\Omega_j}(\boldsymbol{K}\boldsymbol{R}_j), \; j = 1, ..., n_2, \tag{12}$$

where $\boldsymbol{R}_j \overset{\text{i.i.d}}{\sim} \mathcal{N}(\boldsymbol{0}, \sigma^2 \boldsymbol{I}_k)$, $\mathcal{P}_{\Omega_j}(\boldsymbol{K}\boldsymbol{R}_j)$ means keeping the entries in $\Omega_j$ and setting the entries in the complement set of $\Omega_j$ to be zeros, thus $\mathcal{P}_{\Omega_j}(\overline{\boldsymbol{M}}_j)$ has the same set of missing entries as $\mathcal{P}_{\Omega_j}(\boldsymbol{M}_j)$. Note that $\boldsymbol{R}_j$ is stored locally, which are private keys that will NOT be shared with any other node. Then, each node uploads its encrypted vector $\mathcal{P}_{\Omega_j}(\overline{\boldsymbol{M}}_j)$ to the cloud server.

- Third, in lines 2-5 of Alg. 1, after receiving all $n_2$ encrypted vectors $\mathcal{P}_{\Omega_j}(\overline{\boldsymbol{M}}_j)$, $j = 1, ..., n_2$, the server forms an incomplete matrix $\overline{\boldsymbol{M}}_\Omega$ with $\Omega = \bigcup_{j=1,...,n_2} \Omega_j$. Then, the server carries out a matrix completion task in (7), and sends the recovered vector $\overline{\widehat{\boldsymbol{X}}}_j$ back to the $j$-th node, $j = 1, ..., n_2$.

- Finally, in lines 11-13 of Alg. 2, using the locally stored private keys $\boldsymbol{R}_j$, and the public keys $\boldsymbol{K}$, the $j$-th node decrypts its own vector, i.e., $\widehat{\boldsymbol{X}}_j = g^{-1}(\overline{\widehat{\boldsymbol{X}}}_j) = \overline{\widehat{\boldsymbol{X}}}_j - \boldsymbol{K}\boldsymbol{R}_j$, $j = 1, ..., n_2$.

# 4 Homomorphism Property Holds at Price of More Samples

We prove that the homomorphism property holds for the proposed scheme, which guarantees exact recovery on cyphertexts at a cost of more samples. The detailed proofs are given in Appx. C.

**Overview**: Starting from a necessary and sufficient condition in Lemma 1, we relax to a sufficient condition in Lemma 3 for the homomorphism property to hold. Then, we provide a homomorphic version of *Rudelson Selection Estimation Theorem* in Theorem 2 that guarantees Lemma 3 with high probability. Therefore, we obtain a sample complexity for EXACT recovery in Theorem 3, where our interesting finding is that *the homomorphism property holds at price of more samples*.

## 4.1 Sufficient Condition for Low-rank Matrix Completion

We start from a necessary and sufficient condition for low-rank matrix completion. Note that a similar necessary and sufficient condition for sparse vector recovery is discussed in compressive sensing [3, 8, 46]. Here, we apply a similar argument to obtain Lemma 1.

We define a set of matrices with rank at most $r$ and a rank-descent cone as follows

$$\begin{cases} \mathcal{M} = \{\boldsymbol{X} \in \mathbb{R}^{n_1 \times n_2} : \text{rank}(\boldsymbol{X}) \leq r\}, \\ \mathcal{D}_{\mathcal{M}}(\boldsymbol{M}) = \{t(\boldsymbol{X} - \boldsymbol{M}) \in \mathbb{R}^{n_1 \times n_2} : \text{rank}(\boldsymbol{X}) \leq r, \; t \geq 0\}, \end{cases} \tag{13}$$

where $\mathcal{M}$ is the closure of the manifold of rank-$r$ matrices. Accordingly, for $\overline{M}$, we have

$$\begin{cases} \overline{\mathcal{M}} = \{X \in \mathbb{R}^{n_1 \times n_2} : \text{rank}(X) \leq \overline{r}\}, \\ \mathcal{D}_{\overline{\mathcal{M}}}(\overline{M}) = \{t(X - \overline{M}) \in \mathbb{R}^{n_1 \times n_2} : \text{rank}(X) \leq \overline{r}, \ t \geq 0\}. \end{cases} \tag{14}$$

**Lemma 1.** *(**Necessary and sufficient condition for low-rank matrix completion**) $M$ is the unique optimal solution to (6) if and only if $\Omega^\perp \cap \mathcal{D}_{\mathcal{M}}(M) = \{0\}$, where $\Omega^\perp$ denotes $\mathbf{Ker}(\mathcal{P}_\Omega)$.*

**Geometric interpretation**: $M$ is the unique optimal solution to problem (6) if and only if starting from $M$, the rank of $M + D$ increases for all directions $D \in \Omega^\perp$, where $D$ is nonzero. Therefore, the homomorphism property of low-rank matrix completion in problem (7) holds if

$$\Omega^\perp \cap \mathcal{D}_{\mathcal{M}}(M) = \{0\} = \Omega^\perp \cap \mathcal{D}_{\overline{\mathcal{M}}}(\overline{M}). \tag{15}$$

**Lemma 2.** *([15, 7] (Theorem 6.1)) Let $M = U\Sigma V^\top$ be the compact SVD of matrix $M$. The tangent cone $T_{\mathcal{M}}(M)$ of the set $\mathcal{M}$ at $M$ is a linear subspace given by*

$$T_{\mathcal{M}}(M) = \{UA^\top + BV^\top \mid A \in \mathbb{R}^{n_1 \times r}, B \in \mathbb{R}^{n_2 \times r}\} \triangleq T, \tag{16}$$

*and its complementary space is denoted by $T^\perp$.*

Since the rank-descent cone is a subset of the tangent cone defined in (16) ([17], Theorem 4.8), $\mathcal{D}_{\mathcal{M}}(M) \subseteq T$, and $\mathcal{D}_{\overline{\mathcal{M}}}(\overline{M}) \subseteq \overline{T}$, we relax (15) to the following sufficient condition.

**Lemma 3.** *A sufficient condition for the homomorphic property of matrix completion under the proposed scheme in Alg. 1 and Alg. 2 is $\Omega^\perp \cap \overline{T} = \{0\}$.*

**Interpretation**: if $\Omega^\perp \cap \overline{T} = \{0\}$ holds, then we know that $\overline{M} = M + KR$ is the unique optimal solution to problem (7) and $M$ is the unique optimal solution to problem (6). Since $\overline{M} = M + KR$ is a *one-to-one mapping*, a decryption scheme $\overline{M} - KR$ will return the desired true matrix $M$.

## 4.2 Homomorphic Version of Rudelson Selection Estimation Theorem

The Rudelson selection estimation theorem [39] investigates the number of random points needed to bring a convex body into a nearly isotropic position. Such an approximate isometry property is fundamentally useful to characterize the number of entries needed to complete a low-rank matrix.

**Definition 5.** *(Coherence) Let $U \in \mathbb{R}^{n \times r}$ be the $r$ left singular vectors of $M$ (corresponds to the column subspace $\mathcal{S}(M)$) and $P_U$ be the orthogonal projection onto $U$. Then the coherence of $U$ (or $\mathcal{S}(M)$, respectively) is defined as*

$$\mu(\mathcal{S}(M)) = \mu(U) \triangleq \frac{n}{r} \max_{1 \leq i \leq n} ||P_U e_i||_2^2 = \frac{n}{r} \max_{1 \leq i \leq n} ||U^\top e_i||_2^2, \tag{17}$$

*since $(U^\top U)^{-1} = I$ and $U$ is orthonormal.*

The concept "coherence" measures the relationship between a low-dimensional space and the observation operator $\mathcal{P}_\Omega$, namely the cosine (with a scaling factor $\frac{n}{r}$) of the principal angle between the low-dimensional space and a standard basis. $M$ is said to satisfy the *standard incoherence* condition with parameter $\mu_0$ if

$$\mu(U) \leq \mu_0, \quad \text{and} \quad \mu(V) \leq \mu_0. \tag{18}$$

A small $\mu_0$ ensures that the information of the row/column spaces of $M$ is not too concentrated on a small number of rows/columns. It characterizes the contribution of an entry in recovering $M$: a small $\mu_0$ means that each entry provides approximated the same amount of information.

**Theorem 1.** *(**Rudelson selection estimation theorem** [3]) Assume that $\Omega \sim \mathbf{Ber}(p)$ with $p = \Theta(\frac{m}{n_1 n_2})$, and $M$ obeys the standard incoherence condition (18) with parameter $\mu_0$. There is a constant $C_R$ such that for $\beta > 1$, with probability $\geq 1 - 3n_2^{-\beta}$, we have*

$$||p^{-1}\mathcal{P}_T\mathcal{P}_\Omega\mathcal{P}_T - \mathcal{P}_T|| \leq C_R \sqrt{\frac{\mu_0 n_2 r(\beta \log n_2)}{m}} \triangleq \epsilon < 1. \tag{19}$$

We derive the following homomorphic variant of the Rudelson selection estimation theorem [39] and will use it to guarantee Lemma 3. Our new contribution here is to derive the conditions when the approximate isometry property will hold simultaneously for both cyphertexts and plaintexts.

**Theorem 2.** *(**Homomorphic version of Rudelson selection estimation theorem**) Assume that* $\Omega \sim \boldsymbol{Ber}(p)$ *with* $p = \Theta(\frac{m}{n_1 n_2})$, $\boldsymbol{M}$ *and* $\overline{\boldsymbol{M}}$ *satisfy the standard incoherence condition (18) with parameter* $\mu_0$ *and* $\overline{\mu}_0$, *respectively. Under the proposed scheme in Alg. 1 and Alg. 2, there are constants* $C_R, C'_R$ *such that for* $\beta > 1$, *with probability* $\geq 1 - 3n_2^{-\beta}$, *we have*

$$(cyphertext) \ \ ||p^{-1}\mathcal{P}_{\overline{T}}\mathcal{P}_{\Omega}\mathcal{P}_{\overline{T}} - \mathcal{P}_{\overline{T}}|| \leq C'_R\sqrt{\frac{n_2\overline{\mu}_0\overline{r}(\beta \log n_2)}{m}} \triangleq \epsilon' < 1, \ which \ implies$$

$$(plaintext) \ \ ||p^{-1}\mathcal{P}_T\mathcal{P}_{\Omega}\mathcal{P}_T - \mathcal{P}_T|| \leq C_R\sqrt{\frac{n_2\mu_0 r(\beta \log n_2)}{m}} \triangleq \epsilon < 1. \tag{20}$$

Note that $||p^{-1}\mathcal{P}_{\overline{T}}\mathcal{P}_{\Omega}\mathcal{P}_{\overline{T}} - \mathcal{P}_{\overline{T}}|| < 1$ implies that the sufficient condition $\Omega^{\perp} \cap \overline{T} = \{\boldsymbol{0}\}$ holds.

### 4.3 Sample Complexity for EXACT Recovery

Then, we prove Theorem 3 that the homormophism property holds for the proposed scheme in Alg. 1 and Alg. 2, provided that there are sufficient number of observations ($|\Omega|$ is large enough).

**Theorem 3.** *For Alg. 1 and Alg. 2 with probability* $\geq 1 - 3n_2^{-\beta}$, *the homomorphism property holds if* $p \geq \frac{C_0 \overline{\mu}_0 \overline{r}(\beta \log n_2)}{n_1}$, *where* $C_0$ *is a positive constant.*

Next, we characterize the coherence change of $\overline{\mu}_0$ and provide the sample complexity for the EXACT recovery in Alg. 1 and Alg. 2.

**Lemma 4.** *The new coherence under the proposed scheme in Alg. 1 and Alg. 2 satisfies*

$$\overline{\mu}_0 \leq \frac{r}{\overline{r}}\mu_0 + C \max(\frac{k}{\overline{r}}, \frac{\log n_2}{\overline{r}}), \ \ with \ probability \geq 1 - cn_2^{-3}\log n_2. \tag{21}$$

Combining Theorem 3 and Lemma 4, we characterize the required number of entries.

**Corollary 1.** *For Alg. 1 and Alg. 2, with probability* $\geq 1 - 6n_2^{-\beta} - cn_2^{-3}\log n_2$, *the homomorphism property holds if* $p \geq \frac{C_0(r\mu_0 + C\max(k, \log n_2))(\beta \log n_2)}{n_1}$, *where* $c, C_0, C$ *are positive constants.*

## 5 Differential Privacy Property Holds

In this section, we show that the differential privacy holds for the proposed scheme in Alg. 1 and Alg. 2. It is well-known that one can achieve $(\epsilon, \delta)$-differential privacy by adding Gaussian noise.

**Definition 6.** *(**Privacy loss as a random variable** [12]) Considering a mechanism* $\mathcal{A}$ *on a pair of databases* $D, D'$. *For an outcome* $o \in \mathcal{O}$, *the privacy loss on* $o$ *is defined as the logarithmic ratio between the probability to observe* $o$ *on input* $D$ *compared to that on input* $D'$:

$$\mathcal{L}^{(o)}_{\mathcal{A}(D)||\mathcal{A}(D')} = \ln \frac{\mathbb{P}(\mathcal{A}(D) = o)}{\mathbb{P}(\mathcal{A}(D') = o)}, \tag{22}$$

*where* $\mathbb{P}(\mathcal{A}(D) = o)$ *is a probability density over a continuous set* $\mathcal{O}$.

Two potential issues of the proposed scheme in Alg. 1 and Alg. 2 is the projection recovery and the rank value $r$ may be unknown. Namely, for a single-round encryption case, one can do a corresponding projection to obtain the real data. Therefore, we execute the proposed scheme twice and introduce two parameters $\sigma_1$ and $\sigma_2$:

- First-round encryption: the server randomly generats a matrix $\boldsymbol{K}^1 \in \mathbb{R}^{n_1 \times k}$ and each node generates $k$ random numbers $\boldsymbol{R}_j^1 \overset{\text{i.i.d}}{\sim} \mathcal{N}(\boldsymbol{0}, \sigma_1^2 \boldsymbol{I}_k)$. Then, we have $\mathcal{P}_{\Omega}(\overline{\boldsymbol{M}}) = \mathcal{P}_{\Omega}(\boldsymbol{M}) + \mathcal{P}_{\Omega}(\boldsymbol{K}^1 \boldsymbol{R}^1)$.

- Second-round encryption: the server obtains the column space of $\overline{\boldsymbol{M}}$ as $\boldsymbol{K}^2 \in \mathbb{R}^{n_1 \times \overline{r}}$ with $\overline{r} = r + k$ and then each node generates $r + k$ random numbers $\boldsymbol{R}_j^2 \overset{\text{i.i.d}}{\sim} \mathcal{N}(\boldsymbol{0}, \sigma_2^2 \boldsymbol{I}_{r+k})$. Then we have $\mathcal{P}_{\Omega}(\overline{\overline{\boldsymbol{M}}}) = \mathcal{P}_{\Omega}(\overline{\boldsymbol{M}}) + \mathcal{P}_{\Omega}(\boldsymbol{K}^2 \boldsymbol{R}^2)$.

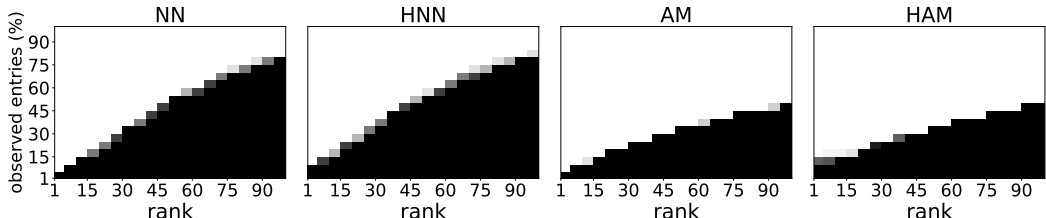

Figure 3: Comparing NN and AM algorithms with their homomorphic versions. The figure plots the success rates within 10 trials, where the white and black cells mean "success" and "fail". The trial is "success" if RSE $\leq 10^{-5}$. We set $k = 10$ in Alg. 1 and Alg. 2.

Theorem 4 states that the proposed scheme satisfies the subspace-aware joint $(\epsilon, \delta)$-differential privacy in Section 2.3.2. The detailed proofs are given in Appx. D, where the key is to quantify $\sigma$ under which the random variable privacy loss in (22) is bounded by $\epsilon$, with probability at least $1 - \delta$.

**Theorem 4.** *Let $\epsilon \in (0,1)$ and $c^2 > 2\ln(1.25/\delta)$. Assume the true data matrix $\boldsymbol{M} \in \mathbb{R}^{n_1 \times n_2}$ has rank $r$ and each column has bounded $\ell_2$-norm, i.e., $\Delta = \max_{1 \leq j \leq n_2} ||\boldsymbol{M}_j||_2 \leq L$. Let $\boldsymbol{R}_j^1 \sim \mathcal{N}(\boldsymbol{0}, \sigma_1^2 \boldsymbol{I}_k)$ with $\sigma_1 \geq 2cL\sqrt{2\ln(2/\delta)}/\epsilon$ and $\boldsymbol{R}_j^2 \sim \mathcal{N}(\boldsymbol{0}, \sigma_2^2 \boldsymbol{I}_{k+r})$ with $\sigma_2 \geq 2cL\sqrt{2\ln(2/\delta)}/\epsilon$, then the encryption and decryption scheme in Alg. 1 and Alg. 2, satisfies the subspace-aware joint $(\epsilon, \delta)$-differential privacy property.*

A substantial improvement is: for the same level of privacy (the same parameters $\epsilon$, $\delta$ in the above joint $(\epsilon, \delta)$-DP property), our algorithms are able to achieve EXACT recovery. Note that by proving the homomorphism property and characterising the sample complexity, we reduce the error bound $O(\sqrt[10]{n_1^3 n_2})$ from [20] to ZERO since we have EXACT recovery.

# 6 Performance Evaluation

We evaluate the proposed scheme on synthetic data and real-world datasets using two matrix completion algorithms [41, 18], verifying the homomorphism property.

## 6.1 Experimental Settings

**Datasets**. We experiment with synthetic data and real-world datasets. The synthetic data is generated randomly according to the low-rank $1,000 \times 1,000$ matrix model and serves as well-controlled inputs for verification. The real-world datasets include two benchmark datasets for recommendation systems, namely the *MovieLens10M (Top 400)*[3] and *Netflix (Top 400)* datasets. The MovieLens dataset contains ratings of 400 most rated movies made by approximately $7,000$ users, and the Netflix dataset contains ratings of 400 most rated movies made by approximately 480 thousand users.

**Matrix completion algorithms**. For the matrix completion on the server, we use nuclear-norm minimization (NN) and alternating minimization (AM) algorithms. In Section 6.2, we compare both algorithms with their homomorphic versions. In Section 6.3, on the real-world datasets, we also include the private Frank-Wolf (FW) algorithm [20] for comparison.

**Performance metric**. We measure the recovery error via the relative square root error RSE $= \frac{||\widehat{M} - M||_F}{||M||_F}$. All experiments are executed for ten times and we report the average results.

## 6.2 Results on Synthetic Data

We experiment with randomly generated low-rank matrices on NN and AM algorithms and their homomorphic versions HNN and HAM. We vary the rank $r$ of the generated matrix and the percentage of observed entries from 1, 5, to 95. As shown in Fig. 6.2, we observe two trends: 1) for a certain rank $r$, the success rate increases as the percentage of observed entries increases; and 2) for a certain percentage of observed entries, the success rate decreases as the rank $r$ increases. On the other hand, we find that the HNN and HAM need slightly more observed entries to reach the success threshold,

---

[3]https://movielens.org/

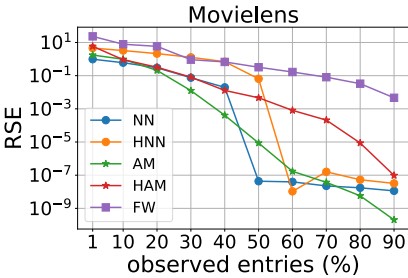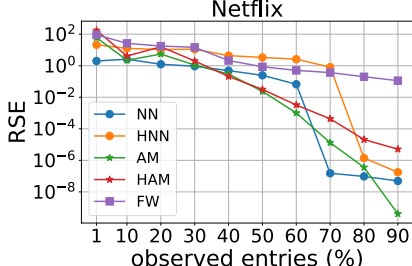

Figure 4: Results on MovieLens10M and Netflix datasets. We vary the percentage of observed entries and measure the RSE recovery error.

which verifies Theorem 3 that the scheme guarantees exact recovery at a cost of more samples. As an interpretation, the homomorphic version is to hide the plaintext matrix into a larger space, namely from rank $r$ to rank $r + k$. In this case, given that we set $k = 10$ for the experiments, we find that the results of HNN and HAM can be obtained by shifting the results of their counterparts left one grid.

### 6.3 Results on MovieLens10M and Netflix Datasets

Fig. 4 shows the results on MovieLens10M and Neflix datasets. For the newly introduced compared algorithm FW, we set the privacy parameter $\epsilon = 2 \log(1/\delta)$ and $\delta = 10^{-6}$. For the NN and AM algorithms, the setting is the same in Section 6.2.

First of all, we observe that the homomorphic algorithms can achieve significantly lower recovery errors than the error of FW algorithm. This points out the difference between the proposed scheme and existing strategies, in which we do not sacrifice the recovery error to improve the privacy. On the other hand, we find that the homomorphic algorithms can reach the same level of recovery error as the vanilla algorithms on plaintexts, but need more samples. Such a performance is consistent with our theoretical proofs and our observations in Section 6.2. Moreover, we analyze the impact of increasing the percentage of observed entries on three types of algorithms, as shown in Fig. 4. For AM and FW algorithms, the recovery error decreases smoothly as the percentage increases (note that the y-axis decreasing in $\log$). However, the NN algorithm demonstrates a significant error drop as we increase the percentage of observed entries.

## 7 Conclusion and Future Work

This work studied the problem of privacy-preserving data completion in a distributed manner. To address the privacy concern, we define the homomorphic matrix completion problem and propose a homomorphic encryption-decryption scheme. Unlike existing works that preserve privacy by sacrificing recovery accuracy, our work guarantees the EXACT recovery while making a tradeoff between privacy and the number of samples. Then, we theoretically prove that the proposed scheme satisfies the homomorphism and differential privacy properties. Experimentally, we show that the proposed scheme is compatible with two matrix completion algorithms, namely the nuclear norm minimization and alternating minimization, and verify the homomorphism property.

In the future, it would be interesting to extend this homomophic framework to the tensor completion problem [31, 32]. It would also be practically interesting to study federated learning application [24] and develop high-performance implementations for high-dimensional data analysis [50, 49, 36, 50].

## Acknowledgement

Xiao-Yang Liu would like to thank Prof. John Wright (Department of Electrical Engineering at Columbia University) for his insightful sharing.

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

## Broader Impact Statement

This paper is within the area of private machine learning, which calls for privacy-preserving data completion by proposing a homomorphic encryption-decryption scheme. Due to the wide application areas of the matrix completion problem, this work may have broad practical impact in recommendation systems, global positioning, system identification and mobile social networks, etc.

