# A Preliminaries

**Notations**: Lower case bold letters denote vectors, while capital bold face letters denote matrices, e.g., $\boldsymbol{x} \in \mathbb{R}^n$ and $\boldsymbol{X} \in \mathbb{R}^{n_1 \times n_2}$. Let $\boldsymbol{e}_i$ denote the $i$-th standard basis. For matrix $\boldsymbol{X}$, its $(i,j)$-th element is $\boldsymbol{X}_{ij}$ or $\boldsymbol{X}(i,j)$ and its $j$-th column is $\boldsymbol{X}_j$. The identity matrix $\boldsymbol{I}$ is a square matrix, sometimes we use $\boldsymbol{I}_n$ to specify its size $n \times n$, and the identity linear operator is denoted by $\mathcal{I}$. The transpose of a vector/matrix is indicated by a superscript $\top$, e.g., $\boldsymbol{x}^\top$ and $\boldsymbol{X}^\top$. The concatenation of two matrices (with the same number of rows), $\boldsymbol{A} \in \mathbb{R}^{n_1 \times n_2}$ and $\boldsymbol{B} \in \mathbb{R}^{n_1 \times n_3}$, is denoted by $[\boldsymbol{A}, \boldsymbol{B}] \in \mathbb{R}^{n_1 \times (n_2 + n_3)}$. By *with high probability* (w.h.p.) we mean that with probability at least $1 - c_1 n^{-c_2}$ for some constants $c_1, c_2 > 0$. Denote the Gaussian distribution by $\mathcal{N}(0, \sigma^2)$, with mean 0 and standard deviation $\sigma$. We use an overline to represent the encrypted version of a variable. Variables before encryption are called *plaintexts*, e.g., $\boldsymbol{X}$, while the encrypted variables are called *cyphertexts*, e.g., $\overline{\boldsymbol{X}}$.

We use an overline to represent the encrypted version of a variable on a cloud server. Variables before encryption are called *plaintexts*, e.g., $\boldsymbol{X}$, while the corresponding encrypted variables are called *cyphertexts*, e.g., $\overline{\boldsymbol{X}}$. Let set $\Omega \subseteq \{(1,1), (1,2), ..., (n_1, n_2)\}$ index the observed entries. We represent the observed entries as $\boldsymbol{M}_\Omega$, while for analysis purpose we define a corresponding linear operator $\mathcal{P}_\Omega : \mathbb{R}^{n_1 \times n_2} \to \mathbb{R}^{n_1 \times n_2}$ to represent the observation model as follows

$$[\mathcal{P}_\Omega(\boldsymbol{M})]_{ij} = (\boldsymbol{M}_\Omega)_{ij} = \begin{cases} \boldsymbol{M}_{ij}, & \text{if } (i,j) \in \Omega \\ 0, & \text{otherwise.} \end{cases} \tag{23}$$

We assume the true matrix $\boldsymbol{M}$ is low-rank, i.e., $\text{rank}(\boldsymbol{M}) = r \ll \min(n_1, n_2)$. The singular value decomposition (SVD) is $\boldsymbol{M} = \boldsymbol{U}\boldsymbol{S}\boldsymbol{V}^\top$, where $\boldsymbol{U} \in \mathbb{R}^{n_1 \times r}$ denotes the $r$ left singular vectors (corresponding to the column subspace), $\boldsymbol{V} \in \mathbb{R}^{n_2 \times r}$ denotes the $r$ right singular vectors (corresponding to the row subspace), and $\boldsymbol{S} = \text{diag}(\sigma_i) \in \mathbb{R}^{r \times r}$ where $\sigma_i$ is the $i$-th largest singular value and $\sigma_1 \geq \sigma_2 \geq \cdots \geq \sigma_r \geq 0$. The nuclear norm of $\boldsymbol{M}$ is $||\boldsymbol{M}||_* = \sum_{i=1}^r \sigma_i$. The $\ell_2$-norm of a vector is $||\boldsymbol{x}||_2$, while the Frobenius norm of a matrix is $||\boldsymbol{M}||_F = \sqrt{\sum_{i,j} |\boldsymbol{M}_{ij}|^2}$. The operator norm (spectral norm) of a matrix is equal to its first singular value, i.e., for $\boldsymbol{M} \in \mathbb{R}^{n_1 \times n_2}$,

$$||\boldsymbol{M}|| \triangleq \sup_{\boldsymbol{x} \in \mathbb{R}^{n_2}, \, ||\boldsymbol{x}||_2 \leq 1} ||\boldsymbol{M}\boldsymbol{x}||_2 = \sigma_1(\boldsymbol{M}), \tag{24}$$

while more generally, the operator norm of a linear operator $\mathcal{L}$ is

$$||\mathcal{L}|| = \sup_{||\boldsymbol{X}||_F \leq 1} ||\mathcal{L}(\boldsymbol{X})||_F. \tag{25}$$

**Definition 7.** *(Column subspace and null space [28]) Let $\boldsymbol{A} \in \mathbb{R}^{n_1 \times n_2}$. The set*

$$\mathcal{S}(\boldsymbol{A}) = \{\boldsymbol{b} \in \mathbb{R}^{n_1} \,|\, \boldsymbol{b} = \boldsymbol{A}\boldsymbol{x}, \, \boldsymbol{x} \in \mathbb{R}^{n_2}\} \tag{26}$$

*is the column space or range of $\boldsymbol{A}$, and the set*

$$\boldsymbol{Ker}(\boldsymbol{A}) = \{\boldsymbol{x} \in \mathbb{R}^{n_2} \,|\, \boldsymbol{A}\boldsymbol{x} = \boldsymbol{0}\} \tag{27}$$

*is the kernel or (right) null space of $\boldsymbol{A}$.*

**Definition 8.** *(Row subspace and null space [28]) Let $\boldsymbol{A} \in \mathbb{R}^{n_1 \times n_2}$. The set*

$$\mathcal{S}(\boldsymbol{A}^\top) = \{\boldsymbol{d} \in \mathbb{R}^{n_2} \,|\, \boldsymbol{d} = \boldsymbol{A}^\top \boldsymbol{y}, \, \boldsymbol{y} \in \mathbb{R}^{n_1}\} \tag{28}$$

*is the row space of $\boldsymbol{A}$, and the set*

$$\boldsymbol{Ker}(\boldsymbol{A}^\top) = \{\boldsymbol{y} \in \mathbb{R}^{n_1} \,|\, \boldsymbol{A}^\top \boldsymbol{y} = \boldsymbol{0}\} \tag{29}$$

*is the left null space of $\boldsymbol{A}$.*

According to Definition 7, the kernel space of the linear operator $\mathcal{P}_\Omega$ is $\boldsymbol{Ker}(\mathcal{P}_\Omega) = \{\boldsymbol{Z} \in \mathbb{R}^{n_1 \times n_2} \,|\, \mathcal{P}_\Omega(\boldsymbol{Z}) = \boldsymbol{0}\}$, which is denoted as $\Omega^\perp$ for simplicity. We adopt the notation $\Omega^\perp$ since $\boldsymbol{Ker}(\mathcal{P}_\Omega)$ equals to a space on the complement set of $\Omega$.

Let $\Omega \sim \boldsymbol{Uni}(m)$ denote a set with $m$ entries, which is sampled uniformly from all sets of $m$ entries, and $\Omega \sim \boldsymbol{Ber}(p)$ denote a set with $\mathbb{E}|\Omega| = m$ entries, each sampled independently according to the Bernoulli model in Definition 9.

**Definition 9.** *(**Bernoulli model** [4]). Let $\{\gamma_{ij}, i = 1, .., n_1, j = 1, ..., n_2\}$ be a sequence of independent identically distributed (i.i.d.) binary Bernoulli random variables with*

$$\mathbb{P}(\gamma_{ij} = 1) = p \triangleq \frac{m}{n_1 n_2}, \quad \mathbb{P}(\gamma_{ij} = 0) = 1 - p. \tag{30}$$

*Correspondingly, we define a set*

$$\Omega = \{(i,j)|\ \gamma_{ij} = 1\} \text{ with } \mathbb{E}|\Omega| = m. \tag{31}$$

**Definition 10.** *(**Random orthogonal model** [4]) The set $\{\boldsymbol{u}_j \in \mathbb{R}^n,\ j = 1, ..., r\}$ is selected uniformly at random among all sets of $r$ orthonormal vectors.*

# B  Formal Definition of Homomorphic Matrix Completion Problem

Now, we define the *homomorphic matrix completion problem*, and further specify the *homomorphism property* in the context of matrix completion.

**Definition 11.** *(**Homomorphic matrix completion (HMC)**) Assume the true matrix $\boldsymbol{M}$ is low-rank with $rank(\boldsymbol{M}) = r \ll \min(n_1, n_2)$, the observed entries $\mathcal{P}_\Omega(\boldsymbol{M}) \in \mathbb{R}^{n_1 \times n_2}$, $\Omega \subseteq [n_1] \times [n_2]$, and let the matrix completion problem (6) be formally expressed as $\widehat{\boldsymbol{X}} = f(\mathcal{P}_\Omega(\boldsymbol{M}), r)$ where $f(\cdot)$ is a mapping[4]. Denote the pair of encryption and decryption schemes[5] by $\{g(\cdot), g^{-1}(\cdot)\}$, the homomorphic matrix completion problem estimates $\widehat{\boldsymbol{X}} = g^{-1}(\widehat{\overline{\boldsymbol{X}}})$, where $\widehat{\overline{\boldsymbol{X}}} = f(\mathcal{P}_\Omega(g(\boldsymbol{M})), \overline{r})$ denotes the following problem*

$$\text{Find a matrix } \overline{\boldsymbol{X}} \in \mathbb{R}^{n_1 \times n_2}, \ \ s.t. \ \ \mathcal{P}_\Omega(\overline{\boldsymbol{X}}) = \mathcal{P}_\Omega(\overline{\boldsymbol{M}}), \ rank(\overline{\boldsymbol{X}}) \leq \overline{r}, \tag{32}$$

*where $\mathcal{P}_\Omega(\overline{\boldsymbol{M}}) = \mathcal{P}_\Omega(g(\boldsymbol{M}))$, and $\overline{r} = rank(\overline{\boldsymbol{M}})$ may be slightly bigger than $r$ (this change is made by the encryption scheme $g(\cdot)$).*

**Definition 12.** *(**The homomorphism property**) With the same assumptions in Def. 11, we say that the homomorphism property holds for the HMC problem if there exists a pair of encryption and decryption schemes $\{g(\cdot), g^{-1}(\cdot)\}$, such that $f(\mathcal{P}_\Omega(\boldsymbol{M}), r) = g^{-1}(f(g(\boldsymbol{M}), \overline{r}))$, and both $f(\mathcal{P}_\Omega(\boldsymbol{M}), r)$ and $f(g(\boldsymbol{M}), \overline{r})$ are functions[6].*

**Remark 1.** *Our HMC problem shares a similar paradigm with the fully homomorphic encryption (FHE) scheme [14, 45], but they are essentially different. The FHE scheme provides an one-to-one mapping between plaintexts and cyphertexts for both the addition and multiplication operations, i.e.,*

$$\begin{cases} a + b = c, \\ a \cdot b = d, \end{cases} \quad g^{-1}(\cdot) \longleftrightarrow g(\cdot), \quad \begin{cases} g(a) + g(b) = g(c), \\ g(a) \cdot g(b) = g(d), \end{cases} \tag{33}$$

*where $g(\cdot)$ and $g^{-1}(\cdot)$ denote the encryption and decryption schemes, respectively. However, throughout the paper, the HMC problem concerns the equivalence between $\widehat{\boldsymbol{X}}$ and $g^{-1}(\widehat{\overline{\boldsymbol{X}}})$, where the recovery of $\overline{\boldsymbol{M}}$ and $\boldsymbol{M}$ are essentially different from the addition and multiplication operations.*

## B.1  Using the Bernoulli Model to Approximate the Uniform Model

If one can bound the probability of failure for the Bernoulli model, then the failure probability for the uniform model will be no more than twice as large, appeared in Section II.C of [5]. For completeness, we formally describe the following two lemmas.

**Lemma 5.** *[5] The probability that the matrix completion problem (6) fails when the set of observed entries is sampled uniformly from the collection of set of size $m$ (i.e., $\Omega \sim \boldsymbol{Uni}(m)$) is less than or equal to twice the probability that (6) fails when $m$ entries are sampled according to the Bernoulli model in Def. 9 (i.e., $\Omega' \sim \boldsymbol{Bern}(p)$ with $p = \frac{m}{n_1 n_2}$).*

---

[4]The mapping $f(\cdot)$ is a function when the recovered $\widehat{\boldsymbol{X}}$ is unique.

[5]Note that we do not use the notations $\boldsymbol{Enc}(\cdot)$ and $\boldsymbol{Dec}(\cdot)$ that are used in the cryptography community, since here we deal with symmetric encryption and decryption schemes, where the decryption scheme is an inverse function of the encryption scheme.

[6]Here, the mapping $f(\cdot)$ for both plaintexts and cyphertexts are functions when the recovered $\widehat{\boldsymbol{X}}$ and $\widehat{\overline{\boldsymbol{X}}}$ are unique, respectively.

*Proof.* The proof follows the argument in Section II.C of [5]. Let $\Omega$ of size $m$ be drawn using the uniform model, and $\Omega'$ be drawn from the Bernoulli model $\Omega' \sim \mathbf{Bern}(p)$ with $p = \frac{m}{n_1 n_2}$). We have

$$\mathbb{P}(\text{Failure}(\Omega')) = \sum_{k=1}^{n_1 n_2} \mathbb{P}(\text{Failure}(\Omega')| \ |\Omega'| = k)\mathbb{P}(|\Omega'| = k)$$
$$= \sum_{k=1}^{n_1 n_2} \mathbb{P}(\text{Failure}(\Omega_k))\mathbb{P}(|\Omega'| = k) \tag{34}$$

where $\Omega_k$ denote a set of entries of size $k$ sampled uniformly at random with $|\Omega_k| = k$.

We observe the following two inequalities:

- $\mathbb{P}(\text{Failure}(\Omega_k))$ is a nonincreasing function of $k$, i.e.,

$$\mathbb{P}(\text{Failure}(\Omega_k)) \geq \mathbb{P}(\text{Failure}(\Omega_{k'})), \quad \text{if } k \leq k'. \tag{35}$$

- Since $m = p \cdot n_1 n_2$ is an integer, it is the median of $|\Omega'|$ [5], i.e.,

$$\mathbb{P}(|\Omega'| \leq m - 1) < \frac{1}{2} \leq \mathbb{P}(|\Omega'| \leq m). \tag{36}$$

.

Then, we have

$$\mathbb{P}(\text{Failure}(\Omega')) \geq \sum_{k=1}^{m} \mathbb{P}(\text{Failure}(\Omega_k))\mathbb{P}(|\Omega'| = k) \geq \mathbb{P}(\text{Failure}(\Omega)) \sum_{k=1}^{m} \mathbb{P}(|\Omega'| = k)$$
$$\geq \frac{1}{2}\mathbb{P}(\text{Failure}(\Omega)). \tag{37}$$

$\square$

**Lemma 6.** *[5] Let $n$ be the number of Bernoulli trials and suppose that $\Omega \sim \mathbf{Ber}(m/n)$. Then, with probability at least $1 - n^{-10}$, $|\Omega| = \Theta(m)$, provided that $m \geq c \log n$ for a constant $c$.*

## C  Proofs for Section 4

### C.1  Proof of Lemma 1

*Proof.* Problem (6) is equivalent to the following problem

$$\text{Find a matrix } \boldsymbol{D} \in \mathbb{R}^{n_1 \times n_2} \quad \text{s.t. } \operatorname{rank}(\boldsymbol{M} + \boldsymbol{D}) \leq r, \ \boldsymbol{D} \in \Omega^{\perp}, \tag{38}$$

where $\boldsymbol{D} \in \Omega^{\perp} = \{\boldsymbol{D} \in \mathbb{R}^{n_1 \times n_2} : \mathcal{P}_{\Omega}(\boldsymbol{M} + \boldsymbol{D}) = \mathcal{P}_{\Omega}(\boldsymbol{M})\}$ (i.e., $\mathcal{P}_{\Omega}(\boldsymbol{D}) = \boldsymbol{0}$)) is equivalent to the constraint $\mathcal{P}_{\Omega}(\boldsymbol{X}) = \mathcal{P}_{\Omega}(\boldsymbol{M})$ in (6).

Then, $\boldsymbol{M}$ is the unique optimal solution to (6) is equivalent to $\boldsymbol{D} = \boldsymbol{0}$. We know that $\operatorname{rank}(\boldsymbol{M} + \boldsymbol{D}) \leq r$ is equivalent to $\boldsymbol{D} \in \mathcal{D}_{\mathcal{M}}(\boldsymbol{M})$ according to (13). Therefore, $\boldsymbol{D} = \boldsymbol{0}$ is equivalent to $\Omega^{\perp} \cap \mathcal{D}_{\mathcal{M}}(\boldsymbol{M}) = \{\boldsymbol{0}\}$, namely, set $\mathcal{D}_{\mathcal{M}}(\boldsymbol{M})$ and set $\Omega^{\perp}$ intersect uniquely at $\boldsymbol{0}$. The lemma is proved.

Interested readers can check a similar proof for Lemma 3.34 (page 116) of [46].

$\square$

### C.2  Proof of Lemma 3

We provide a geometric interpretation of the homomorphism property in the context of matrix completion problems.

- The homomorphism property under the scheme in Alg. 1 and Alg. 2 can be mathematically interpreted as: $\boldsymbol{M}$ is the unique optimal solution to problem (6), while simultaneously $\boldsymbol{M} + \boldsymbol{K}\boldsymbol{R}$ is the unique optimal solution to problem (7). According to Lemma 1, the necessary and sufficient condition would be $\Omega^{\perp} \cap \mathcal{D}_{\mathcal{M}}(\boldsymbol{M}) = \{\boldsymbol{0}\}$ and $\Omega^{\perp} \cap \mathcal{D}_{\overline{\mathcal{M}}}(\overline{\boldsymbol{M}}) = \{\boldsymbol{0}\}$.

- Since it always holds that $\mathcal{D}_{\mathcal{M}}(\boldsymbol{M}) \subseteq T$ and $\mathcal{D}_{\overline{\mathcal{M}}}(\overline{\boldsymbol{M}}) \subseteq \overline{T}$ according to Lemma 7 and Lemma 8, a sufficient condition for the homomorphic property is that $\Omega^{\perp} \cap \overline{T} = \{\boldsymbol{0}\}$ and $\Omega^{\perp} \cap T = \{\boldsymbol{0}\}$ hold simultaneously. The scheme in Alg. 1 and Alg. 2 satisfies that $T \subseteq \overline{T}$, thus $\Omega^{\perp} \cap \overline{T} = \{\boldsymbol{0}\}$ implies $\Omega^{\perp} \cap T = \{\boldsymbol{0}\}$.

- If the sampling probability $p$ is large enough, with high probability, it is guaranteed that $\Omega^{\perp} \cap \overline{T} = \{\boldsymbol{0}\}$, which also implies $\Omega^{\perp} \cap T = \{\boldsymbol{0}\}$ since $T \subseteq \overline{T}$.

First, according to Lemma 7 and Lemma 8, we have

$$\mathcal{D}_{\mathcal{M}}(\boldsymbol{M}) = \mathbf{cone}\{\mathcal{M} - \{\boldsymbol{M}\}\} \subseteq T, \ \ \mathcal{D}_{\overline{\mathcal{M}}}(\overline{\boldsymbol{M}}) = \mathbf{cone}\{\overline{\mathcal{M}} - \{\overline{\boldsymbol{M}}\}\} \subseteq \overline{T}. \tag{39}$$

Lemma 7 states that the tangent cone of the set $\mathcal{M}$ evaluated at $\boldsymbol{M}$ has a closed-form expression, namely, it is just the space $T$ in (4). Lemma 8 claims that the tangent cone of the set $\mathcal{M}$ evaluated at $\boldsymbol{M}$ is slightly larger than the cone $\mathbf{cone}(\mathcal{M} - \{\boldsymbol{M}\})$, where $\{\boldsymbol{M}\}$ denotes a set.

**Definition 13.** *([40]) A matrix $\Xi \in \mathbb{R}^{n_1 \times n_2}$ is tangent to $\mathcal{M}$ at $\boldsymbol{M}$ if there exists a sequence $\{\boldsymbol{M}^n\}$ contained in $\mathcal{M}$ and converging to $\boldsymbol{M}$, and a sequence $\{a_n\}$ of nonnegative numbers, such that the sequence $a_n(\boldsymbol{M}^n - \boldsymbol{M})$ converges to $\Xi$. Then, the tangent cone of the set $\mathcal{M}$ at point $\boldsymbol{M}$, denoted by $T_{\mathcal{M}}(\boldsymbol{M})$, is formally defined as follows*

$$T_{\mathcal{M}}(\boldsymbol{M}) = \{\Xi \in \mathbb{R}^{n_1 \times n_2} \mid \exists \boldsymbol{M}^n \subseteq \mathcal{M}, \{a_n\} \subseteq \mathbb{R}^+, \ s.t. \ \boldsymbol{M}^n \to \boldsymbol{M}, a_n(\boldsymbol{M}^n - \boldsymbol{M}) \to \Xi\}. \tag{40}$$

**Lemma 7.** *[7] (Theorem. 6.1)) Let $\boldsymbol{M} = \boldsymbol{U}\Sigma\boldsymbol{V}^{\top}$ be the skinny SVD (or reduced SVD) of matrix $\boldsymbol{M}$. The tangent cone $T_{\mathcal{M}}(\boldsymbol{M})$ of the set $\mathcal{M} = \{\boldsymbol{X} \in \mathbb{R}^{n_1 \times n_2} : \operatorname{rank}(\boldsymbol{X}) \leq r\}$ at $\boldsymbol{M}$ is a linear subspace given by*

$$T_{\mathcal{M}}(\boldsymbol{M}) = \{\boldsymbol{U}\boldsymbol{A}^{\top} + \boldsymbol{B}\boldsymbol{V}^{\top} \mid \boldsymbol{A} \in \mathbb{R}^{n_1 \times r}, \boldsymbol{B} \in \mathbb{R}^{n_2 \times r}\} \triangleq T, \tag{41}$$

*where the complementary space is denoted by $T^{\perp}$.*

**Lemma 8.** *([17], Theorem 4.8) Let $\mathcal{M}$ be a non-empty subset of a real normed space. If $\mathcal{M}$ is star-shaped w.r.t. some $\boldsymbol{M} \in \mathcal{M}$, i.e., $t(\mathcal{M} - \{\boldsymbol{M}\}) \subseteq \mathcal{M} - \{\boldsymbol{M}\}$ for all $t \in [0, 1]$, then if follows*

$$\mathcal{D}_{\mathcal{M}}(\boldsymbol{M}) = \boldsymbol{cone}(\mathcal{M} - \{\boldsymbol{M}\}) \subseteq T_{\mathcal{M}}(\boldsymbol{M}). \tag{42}$$

Second, we point out that the scheme in Alg. 1 and Alg. 2 satisfies $T \subseteq \overline{T}$. It leads to our argument that $\Omega^{\perp} \cap \overline{T} = \{0\}$ implies $\Omega^{\perp} \cap T = \{0\}$, since $0$ belongs to $\Omega^{\perp}$, $\overline{T}$ and $T$ by definition. This condition can be interpreted as follows: with the same observation set $\Omega$, recovering $\overline{M} = M + KR$ (a matrix in a larger subspace) is more difficult than recovering $M$. This is formulated in Lemma 3.

## C.3 Proof of Theorem 2

*Proof.* Applying Theorem 1 for cyphertext $\overline{M}$, then we obtain that with probability at least $1 - 3n_2^{-\beta}$,

$$||p^{-1}\mathcal{P}_{\overline{T}}\mathcal{P}_{\Omega}\mathcal{P}_{\overline{T}} - \mathcal{P}_{\overline{T}}|| \leq C_R' \sqrt{\frac{n_2 \overline{\mu}_0 \overline{r}(\beta \log n_2)}{m}} \triangleq \epsilon', \tag{43}$$

provided that $\epsilon' < 1$.

Then, under the proposed scheme in Alg. 1 and Alg. 2, we show that $\epsilon' < 1$ implies $\epsilon < 1$, and we have $\overline{M} = M + KR$ and $\overline{r} > r$.

Combining the coherence change in Section 4.3, we can easily know that $\mu_0 < \overline{\mu}_0$. Since $r < \overline{r} = \text{rank}(\overline{M})$, we obtain that $\mu_0 r < \overline{\mu r}$. Therefore, there are constants $C_R, C_R'$ such that for $\beta > 1$, $\epsilon' < 1$ implies $\epsilon < 1$. The proof is completed. $\square$

## C.4 Proof of Theorem 3

*Proof.* Recall that Lemma 3 states that a sufficient condition for the homomorphic property is $\Omega^{\perp} \cap \overline{T} = \{0\}$. We use Theorem 2 to derive a sufficient condition $p \geq \frac{C_0 \overline{\mu}_0 \overline{r}(\beta \log n_2)}{n_1}$.

According to Theorem 2, if $\Omega \sim \textbf{Ber}(p)$, with probability at least $1 - 3n_2^{-\beta}$, we have

$$||\mathcal{P}_T - p^{-1}\mathcal{P}_T\mathcal{P}_{\Omega}\mathcal{P}_T|| \leq \epsilon < 1, \quad ||\mathcal{P}_{\overline{T}} - p^{-1}\mathcal{P}_{\overline{T}}\mathcal{P}_{\Omega}\mathcal{P}_{\overline{T}}|| \leq \epsilon' < 1, \tag{44}$$

provided that $p \geq \frac{C_0 \overline{\mu}_0 \overline{r}(\beta \log n_2)}{n_1}$.

Note that $\mathcal{I} = \mathcal{P}_{\Omega} + \mathcal{P}_{\Omega^{\perp}}$, where $\mathcal{P}_{\Omega^{\perp}}$ denotes the projection onto $\Omega^{\perp}$, then

$$\mathcal{P}_{\overline{T}} - p^{-1}\mathcal{P}_{\overline{T}}\mathcal{P}_{\Omega}\mathcal{P}_{\overline{T}} = \mathcal{P}_{\overline{T}} - p^{-1}\mathcal{P}_{\overline{T}}(\mathcal{I} - \mathcal{P}_{\Omega^{\perp}})\mathcal{P}_{\overline{T}} = p^{-1}(\mathcal{P}_{\overline{T}}\mathcal{P}_{\Omega^{\perp}}\mathcal{P}_{\overline{T}} - (1 - p)\mathcal{P}_{\overline{T}}), \tag{45}$$

then by the triangle inequality of the operator norm, we obtain that

$$||\mathcal{P}_{\overline{T}}\mathcal{P}_{\Omega^{\perp}}\mathcal{P}_{\overline{T}}|| \leq p||\mathcal{P}_{\overline{T}} - p^{-1}\mathcal{P}_{\overline{T}}\mathcal{P}_{\Omega}\mathcal{P}_{\overline{T}}|| + (1 - p)||\mathcal{P}_{\overline{T}}|| < \epsilon' p + (1 - p), \tag{46}$$

where we have $||\mathcal{P}_{\overline{T}}|| = 1$ according to Eq. 24, since for $\overline{X} \in \overline{T}$, we have $||\mathcal{P}_{\overline{T}}(\overline{X})|| = \overline{X}$.

Since $||\mathcal{P}_{\Omega^{\perp}}\mathcal{P}_{\overline{T}}||^2 = ||\mathcal{P}_{\overline{T}}\mathcal{P}_{\Omega^{\perp}}\mathcal{P}_{\Omega^{\perp}}\mathcal{P}_{\overline{T}}|| \leq ||\mathcal{P}_{\overline{T}}\mathcal{P}_{\Omega^{\perp}}\mathcal{P}_{\overline{T}}||$, then $||\mathcal{P}_{\Omega^{\perp}}\mathcal{P}_{\overline{T}}|| < \sqrt{1 - p + \epsilon' p} < 1$. Note that $||\mathcal{P}_{\Omega^{\perp}}\mathcal{P}_{\overline{T}}|| < 1$ implies that $\Omega^{\perp} \cap \overline{T} = \{0\}$, which holds by contradiction: if there exists a nonzero matrix $X \in \Omega^{\perp} \cap \overline{T}$, then $||\mathcal{P}_{\Omega^{\perp}}\mathcal{P}_{\overline{T}}(X)|| = X$ and $||\mathcal{P}_{\Omega^{\perp}}\mathcal{P}_{\overline{T}}|| = 1$.

We have shown that if $p \geq \frac{C_0 \overline{\mu}_0 \overline{r}(\beta \log n_2)}{n_1}$, with probability at least $1 - 6n_2^{-\beta}$, it holds for $\Omega^{\perp} \cap \overline{T} = \{0\}$. Combining Lemma 3, the proof is completed. $\square$

## C.5 Proof of Lemma 4

We provide key lemmas of coherence change for the union of two subspaces, which will be used to characterize the coherence change under the scheme in Alg. 1 and Alg. 2.

The concept "coherence" measures the relationship between a low-dimensional space and the observation operator $\mathcal{P}_{\Omega}$, namely the cosine (with a scaling factor $\frac{n}{r}$) of the principal angle between the low-dimensional space and a standard basis.

**Definition 14.** *(Coherence [6]) Let $U \in \mathbb{R}^{n \times r}$ be the $r$ left singular vectors of $M$ (corresponds to the column subspace $\mathcal{S}(M)$) and $P_U$ be the orthogonal projection onto $U$. Then the coherence of $U$ (or $\mathcal{S}(M)$, respectively) is defined as*

$$\mu(\mathcal{S}(M)) = \mu(U) \triangleq \frac{n}{r} \max_{i=1,\ldots,n} ||P_U e_i||_2^2 = \frac{n}{r} \max_{i=1,\ldots,n} ||U(U^{\top}U)^{-1}U^{\top}e_i||_2^2 = \frac{n}{r} \max_{i=1,\ldots,n} ||U^{\top}e_i||_2^2, \tag{47}$$

*since $(U^{\top}U)^{-1} = I$ and $U$ is orthonormal.*

**Remark 2.** *Note that the left singular vectors $U$ (or right singular vectors $V$, respectively) are not uniquely defined and all candidates are unitary transform to each other, i.e., $M = (UW)S(VW)^\top = USV^\top$ where $W \in \mathbb{R}^{r \times r}$ is an orthonormal matrix, i.e., $WW^\top = W^\top W = I$. The coherence of the column subspace $\mu(\mathcal{S}(M))$ (or the row subspace $\mu(\mathcal{S}(M^\top))$, respectively) is uniquely defined, since the coherence is invariant under unitary transformation. Specifically, given two orthonormal matrices $W \in \mathbb{R}^{r \times r}$ and $U \in \mathbb{R}^{n_1 \times r}$, then $\mu(UW) = \mu(U)$, since $||P_{(UW)} \cdot e_i||_2 = ||UW((UW)^\top UW)^{-1}(UW)^\top e_i||_2 = ||U^\top e_i||_2$. Therefore, we will use $\mu(U)$ instead of $\mu(\mathcal{S}(M))$.*

**Lemma 9.** *(**Incoherence change under the union operation**) Let $r = r_1 + r_2$, the matrix $U = [U_1, U_2] \in \mathbb{R}^{n \times (r_1 + r_2)}$ be the concatenation of two non-overlapping orthonormal matrices $U_1 \in \mathbb{R}^{n \times r_1}$ and $U_2 \in \mathbb{R}^{n \times r_2}$. Then, we have $\mu(U) \leq \frac{r_1}{r}\mu(U_1) + \frac{r_2}{r}\mu(U_2)$, and it applies to the union of two subspaces, i.e., $\mu(\mathcal{S}(U)) \leq \frac{r_1}{r}\mu(\mathcal{S}(U_1)) + \frac{r_2}{r}\mu(\mathcal{S}(U_2))$.*

*Proof.* From Def. 14, we have

$$
\begin{aligned}
\mu(U) &= \frac{n}{r}\max_{i=1}^{n} ||[U_1, U_2]^\top e_i||_2^2 = \frac{n}{r}\max_{i=1}^{n} \left( ||U_1^\top(:,i)||_2^2 + ||U_2^\top(:,i)||_2^2 \right) \\
&\leq \frac{n}{r}\max_{i=1}^{n} ||U_1^\top(:,i)||_2^2 + \frac{n}{r}\max_{j=1}^{n} ||U_2^\top(:,j)||_2^2 \\
&\leq \frac{r_1}{r}\mu(U_1) + \frac{r_2}{r}\mu(U_2),
\end{aligned}
\tag{48}
$$

where $\mu(U_1)$ and $\mu(U_2)$ are given in Def. 14. For the union of subspaces, one can easily verify the incoherence change according to Remark 2. $\square$

**Definition 15.** *(**Random orthogonal model** [6]) The set $\{u_j \in \mathbb{R}^n, \ j = 1, ..., r\}$ is selected uniformly at random among all sets of $r$ orthonormal vectors.*

**Lemma 10.** *(**Random subspaces span incoherence subspaces** [6]) Set $r' = \max(r, \log n)$. There exist positive constants $C$ and $c$, the random orthogonal model in Def. 15 obeys*

$$
\max_i ||\mathcal{P}_U e_i||_2^2 = \max_i ||U^\top e_i||_2^2 \leq Cr'/n, \ \text{which yields } \mu(U) \leq C \cdot \max(1, \frac{\log n}{r}), \tag{49}
$$

*with probability $\geq 1 - cn^{-3}\log n$.*

Here we formally prove Lemma 4.

*Proof.* The scheme in Alg. 1 and Alg. 2 can be represented as $\overline{M} = M + KR$. Let $M = USV^\top$ and $\overline{M} = \overline{U}\ \overline{S}\ \overline{V}^\top$, we have the column subspaces $\mathcal{S}(M) = \mathcal{S}(U)$, $\mathcal{S}(\overline{M}) = \mathcal{S}(\overline{U})$, and the row subspaces $\mathcal{S}(M^\top) = \mathcal{S}(V)$, $\mathcal{S}(\overline{M}^\top) = \mathcal{S}(\overline{V})$. Let $KR = U'S'(V')^\top$. Since $K$ and $R$ are randomly generated matrices, thus we know that $U'$ and $V'$ obey the random orthogonal model in Def. 15.

First, with probability 1, we have

$$
\mathcal{S}(U) \subseteq \mathcal{S}(\overline{U}), \ \text{specifically, } \mathcal{S}(\overline{U}) = \mathcal{S}(U) \cup \mathcal{S}(K) = \mathcal{S}(U) \cup \mathcal{S}(U'), \tag{50}
$$

$$
\mathcal{S}(V) \subseteq \mathcal{S}(\overline{V}), \ \text{specifically, } \mathcal{S}(\overline{V}) = \mathcal{S}(V) \cup \mathcal{S}(R^\top) = \mathcal{S}(V) \cup \mathcal{S}(V'). \tag{51}
$$

Secondly, we bound the coherence changes. Note that with probability 1, $\text{rank}(\overline{M}) = \overline{r} = r + k$. From (50), we know that $\overline{U} \in \mathbb{R}^{n_1 \times \overline{r}}$ can be represented as the union of $U \in \mathbb{R}^{n_1 \times r}$ and $U' \in \mathbb{R}^{n_1 \times k}$, i.e., $\overline{U} = [U\ U']$, where $U'$ obeys the random orthogonal model and the non-overlapping requirement in Lemma 9. From Lemma 10, we know that $\mu(U') \leq C \cdot \max(1, \frac{\log n_1}{k})$.

According to Lemma 9, we then obtain that

$$
\mu(\overline{U}) \leq \frac{r}{\overline{r}}\mu_0 + \frac{k}{\overline{r}} \cdot C\max(1, \frac{\log n_1}{k}) \leq \frac{r}{\overline{r}}\mu_0 + C\max(\frac{k}{\overline{r}}, \frac{\log n_1}{\overline{r}}). \tag{52}
$$

Similarly, for the row subspace, we have

$$
\mu(\overline{V}) \leq \frac{r}{\overline{r}}\mu_0 + C\max(\frac{k}{\overline{r}}, \frac{\log n_2}{\overline{r}}), \tag{53}
$$

Therefore,

$$\overline{\mu}_0 \leq \frac{r}{\overline{r}}\mu_0 + C\max(\frac{k}{\overline{r}}, \frac{\log n_2}{\overline{r}}). \tag{54}$$

$\square$

# D Proof for Section 5

## D.1 Proof of Theorem 4

A potential issue of the proposed scheme in Alg. 1 and Alg. 2 is the projection recovery. Namely, for a single-round encryption case, one can do a corresponding projection to obtain the real data. Therefore, we execute the proposed scheme twice and introduce two parameters $\sigma_1$ and $\sigma_2$ in Theorem 4:

- First-round encryption: we randomly generate a matrix $\boldsymbol{K}^1 \in \mathbb{R}^{n_1 \times k}$ and $k$ random numbers $\boldsymbol{R}^1 \overset{\text{i.i.d}}{\sim} \mathcal{N}(\boldsymbol{0}, \sigma_1^2 \boldsymbol{I}_k)$. Then we obtain $\mathcal{P}_\Omega(\overline{M}) = \mathcal{P}_\Omega(M) + \mathcal{P}_\Omega(\boldsymbol{K}^1\boldsymbol{R}^1)$.

- Second-round encryption: we obtain the column space of $\overline{M}$ as $\boldsymbol{K}^2 \in \mathbb{R}^{n_1 \times k}$ and $k$ random numbers $\boldsymbol{R}^2 \overset{\text{i.i.d}}{\sim} \mathcal{N}(\boldsymbol{0}, \sigma_2^2 \boldsymbol{I}_k)$. Then we obtain $\mathcal{P}_\Omega(\overline{\overline{M}}) = \mathcal{P}_\Omega(\overline{M}) + \mathcal{P}_\Omega(\boldsymbol{K}^2\boldsymbol{R}^2)$.

Overview of our proof strategy: 1). we first prove Theorem 5 for a single-round encryption case; 2). Theorem 4 can be easily derived by applying Theorem 5 twice.

**Theorem 5.** *(Gaussian mechanism [12]) Given an arbitrary $n$-dimensional function $\mathcal{A}: \mathcal{T}^n \to \mathcal{O}^n$, define its $\ell_2$-norm sensitivity as $\Delta = \max_{dist(D,D')=1} ||\mathcal{A}(D) - \mathcal{A}(D')||_2$ where $dist(D, D') = 1$ means that $D, D' \in \mathcal{T}^n$ differ from each other by one data entry. The Gaussian mechanism with parameter $\sigma$ adds Gaussian noise $\boldsymbol{w} \overset{\text{i.i.d}}{\sim} \mathcal{N}(\boldsymbol{0}, \sigma^2 \boldsymbol{I}_n)$ to the output. Let $\epsilon \in (0, 1)$ and $c^2 > 2\ln(1.25/\delta)$, then the Gaussian mechanism, i.e., $\mathcal{A}(D) + \boldsymbol{w}$, with parameter $\sigma \geq c\Delta\sqrt{2\ln(2/\delta)}/\epsilon$ satisfies $(\epsilon, \delta)$-differential privacy.*

*Proof.* Let $\boldsymbol{v} \in \mathcal{O}^n$ be a vector with $||\boldsymbol{v}||_2 \leq \Delta$. For a pair of databases $D, D' \in \mathcal{T}^n$ with $dist(D, D') = 1$, we would like to obsecure the vector $\boldsymbol{v} = \mathcal{A}(D) - \mathcal{A}(D')$ by adding Gaussian noise $\boldsymbol{w} \overset{\text{i.i.d}}{\sim} \mathcal{N}(\boldsymbol{0}, \sigma^2 \boldsymbol{I}_n)$, i.e., $\mathcal{A}(D) + \boldsymbol{w}$. Thus an adversary user will not be able to differentiate $D$ from $D'$ by observing $\mathcal{A}(D) + \boldsymbol{w}$ or $\mathcal{A}(D') + \boldsymbol{w}'$. According to the joint $(\epsilon, \delta)$-differential privacy in Def. 3, the key is to quantify $\sigma$ under which the random variable *privacy loss* in (22) is bounded by $\epsilon$, with probability at least $1 - \delta$. Explicitly, we would like to show that

$$\left| \mathcal{L}^{(o)}_{\mathcal{M}(D)||\mathcal{M}(D')} \right| = \left| \ln \frac{\mathbb{P}(\mathcal{A}(D) = o)}{\mathbb{P}(\mathcal{A}(D') = o)} \right| = \left| \ln \frac{\exp(-\frac{1}{2\sigma^2}||o - \mathcal{A}(D)||_2^2)}{\exp(-\frac{1}{2\sigma^2}||o - \mathcal{A}(D')||_2^2)} \right|$$
$$= \left| \ln \frac{\exp(-\frac{1}{2\sigma^2}||\boldsymbol{w}||_2^2)}{\exp(-\frac{1}{2\sigma^2}||\boldsymbol{w} + \boldsymbol{v}||_2^2)} \right| = \left| \frac{1}{2\sigma^2}(||\boldsymbol{w}||_2^2 - ||\boldsymbol{w} + \boldsymbol{v}||_2^2) \right| \leq \epsilon \tag{55}$$

holds with probability at least $1 - \delta$.

Note that $\boldsymbol{w} \overset{\text{i.i.d}}{\sim} \mathcal{N}(\boldsymbol{0}, \sigma^2 \boldsymbol{I}_n)$, combining the fact that *the distribution of a spherically symmetric normal is independent of the orthogonal basis from which its constituent normals are drawn*, we choose to work in a basis that is aligned with $\boldsymbol{v}$. For a basis $\{\boldsymbol{B}_1, \boldsymbol{B}_2, ..., \boldsymbol{B}_n\}$ (or alternatively represented as a matrix $\boldsymbol{B} \in \mathbb{R}^{n \times n}$) with $\boldsymbol{B}_1$ being parallel to $\boldsymbol{w}$, we draw $\boldsymbol{w}$ by drawing signed scalars $\lambda_i \sim \mathcal{N}(0, \sigma^2)$ for $i = 1, ..., n$, then defining $\boldsymbol{w}^i = \lambda_i \boldsymbol{B}_i \in \mathbb{R}^n$ and $\boldsymbol{w} = \sum_{i=1}^{n} \boldsymbol{w}^i \in \mathbb{R}^n$.

Consider the right triangle with base $\boldsymbol{v} + \boldsymbol{w}^i$ and edge $\sum_{i=2}^{n} \boldsymbol{w}^i \in \mathbb{R}^n$, its hypotenuse is $\boldsymbol{v} + \boldsymbol{w}$. Then, we have

$$||\boldsymbol{w}||_2^2 = \sum_{i=1}^{n} ||\boldsymbol{w}^i||_2^2,$$

$$||\boldsymbol{v} + \boldsymbol{w}||_2^2 = ||\boldsymbol{v} + \boldsymbol{w}^1||_2^2 + \sum_{i=2}^{n} ||\boldsymbol{w}^i||_2^2 = (||\boldsymbol{v}||_2 + \lambda_1)^2 + \sum_{i=2}^{n} ||\boldsymbol{w}^i||_2^2, \tag{56}$$

$$||\boldsymbol{v} + \boldsymbol{w}||_2^2 - ||\boldsymbol{w}||_2^2 = ||\boldsymbol{v}||_2^2 + 2\lambda_1||\boldsymbol{v}||_2,$$

where we used the fact $\boldsymbol{B}_1$ is parallel to $\boldsymbol{v}$ and $||\boldsymbol{w}^1||_2^2 = \lambda_1^2$.

Recall that $||\boldsymbol{v}||_2 \leq \Delta$ and $\lambda_i \sim \mathcal{N}(0, \sigma^2)$ for $i = 1, ..., n$, the bound in (55) becomes

$$\left| \ln \frac{\exp(-\frac{1}{2\sigma^2}||\boldsymbol{w}||_2^2)}{\exp(-\frac{1}{2\sigma^2})||\boldsymbol{w}+\boldsymbol{v}||_2^2} \right| = \left| \frac{1}{2\sigma^2}(||\boldsymbol{w}||_2^2 - ||\boldsymbol{w}+\boldsymbol{v}||_2^2) \right| \leq \left| \frac{1}{2\sigma^2}(2\lambda_1\Delta + \Delta^2) \right| \leq \epsilon. \quad (57)$$

The quantity in (57) is bounded by $\epsilon$ whenever $|\lambda_1| < \frac{\sigma^2\epsilon}{\Delta} - \frac{\Delta}{2}$.

Set $\sigma = c\Delta/\epsilon$ and $c^2 > 2\ln(1.25/\delta)$. Let us partition $\mathbb{R}$ into $\mathbb{R} = R_1 \cup R_2$, where $R_1 = \{\lambda \in \mathbb{R} \mid |\lambda| \leq c\Delta/\epsilon\}$ and $R_2 = \{\lambda \in \mathbb{R} \mid |\lambda| > c\Delta/\epsilon\}$, and define $\mathcal{O} = O_1 \cup O_2$ where

$$\begin{aligned} O_1 &= \{\mathcal{A}(x) + \lambda \mid \lambda \in R_1\}, \\ O_2 &= \{\mathcal{A}(x) + \lambda \mid \lambda \in R_2\}. \end{aligned} \quad (58)$$

To ensure the privacy loss is bounded by $\epsilon$ with probability at least $1 - \delta$, one requires

$$\mathbb{P}\left[|\lambda| \geq \frac{\sigma^2\epsilon}{\Delta} - \frac{\Delta}{2}\right] < \delta, \quad (59)$$

or alternatively, one requires

$$\mathbb{P}\left[\lambda \geq \frac{\sigma^2\epsilon}{\Delta} - \frac{\Delta}{2}\right] < \delta/2. \quad (60)$$

Taking the integration of (57), one obtains that

$$\begin{aligned} \mathbb{P}[\mathcal{A}(x) + \lambda \in O] &= \mathbb{P}[\mathcal{A}(x) + \lambda \in O_1] + \mathbb{P}[\mathcal{A}(x) + \lambda \in O_2] \\ &\leq \mathbb{P}[\mathcal{A}(x) + \lambda \in O_1] + \delta \\ &\leq e^\epsilon \cdot \mathbb{P}[\mathcal{A}(y) + \lambda \in O_1] + \delta, \end{aligned} \quad (61)$$

yielding the joint $(\epsilon, \delta)$-differntial privacy for the Gaussian mechanism. $\quad \square$