# OpenReview forum: "Homomorphic Matrix Completion"
_NeurIPS.cc/2022/Conference — NeurIPS 2022 Accept_

### Official Review · Reviewer_WYT8 · 2022-07-11

**Rating:** 7
**Confidence:** 3
**Soundness:** 3 good
**Presentation:** 2 fair
**Contribution:** 4 excellent

**Summary:**

The authors derive a novel homomorphic matrix completion algorithm with a proof that the homomorphism property holds provided certain technical conditions are satisfied, including a probabilistic bound on the number of observed entries required.
They also prove that the novel algorithm satisfies differential privacy constraints.
The authors' scheme solves the matrix completion problem on the server with homomorphically encrypted matrix entries while employing a higher rank constraint using any standard matrix completion method. The proof for the homomorphism property relies upon a homomorphic version of the Rudelson selection estimation theorem from [3].

Experimental results on the Netflix and MovieLens datasets indicate that the homomorphic counterparts of nuclear norm (NN) minimization, dubbed HNN and alternating minimization (AM), dubbed HAM, are only slightly worse than the original ones and that the new schemes outperforms the differentially private Frank Wolf (FW) scheme.


**Questions:**

The parameter $k$ is a design parameter of the algorithm but the authors simply set it to $10$ in all experiments.
Shouldn't it vary with rank $r$ or $p$ ?


**Limitations:**

The limitations of the proposed method seems to be the trade-off between accuracy and privacy / number of samples, as discussed in the Conclusion.

**Strengths And Weaknesses:**

Strengths

1. The theoretical guarantees in the papers as summarized above appear to be quite strong.

2. The experimental results as summarized above appear to be quite solid. The observation regarding the significant drop in error for HNN in Sec. 6.3 is also quite interesting.

3. The authors provide geometric intuition for Lemma 1 regarding why $M$ is the exact solution to Eq (3) when a certain condition holds. It appears that Lemma 1 and this geometric interpretation is original and not already present in reference [3] or [6].

4.
Homomorphic encryption is generally supposed to be slow, but the proposed encryption / decryption method using public / private random matrices seems to be quite fast.

Weaknesses

1. This reviewer found the math in the paper hard to follow and hard to constantly refer to [3] or other references because the paper didn't seem to be self-contained.
The tangent cone mentioned after eq (12) is not defined in the paper and neither is the incoherence parameter used in (13).

2. Both constants $C$ and $c$ are present in Lemma 3 and Corollary 1. It seems to this reviewer that $c = C$.

3. The parameter $\zeta$ in Theorem 4 doesn't seem to be defined anywhere.

4. Typo: "Rank-decent" should be "Rank-descent" after eq (12).

---

> ### Author Response · Authors · 2022-08-03
> **Response to Reviewer WYT8 (1/2)**
>
> Thank you for your positive and thoughtful comments. We would like to address your concerns and answer your questions in the following.
>
> > The experimental results as summarized above appear to be quite solid. The observation regarding the significant drop in error for HNN in Sec. 6.3 is also quite interesting.
>
> Such a significant drop in error is closely related to the phase transition phenomena of matrix completion. It is worth more effort to investigate. We are planning to provide more numerical results in an appendix, for completeness purposes.
>
> > The authors provide geometric intuition for Lemma 1 regarding why $M$ is the exact solution to Eq (3) when a certain condition holds. It appears that Lemma 1 and this geometric interpretation is original and not already present in reference [3] or [6].
>
> Both Lemma 1 and the geometric interpretation are not from [3, 6], and yes, they can be considered original. But we would like to credit them back to [3, 6], since the fundamental ingredients come from [3, 6], through some fruitful discussion with those authors.
>
> Also, Theorem 2 is a homomorphic version of the Rudelson selection estimation theorem. The proof of Theorem 3 in Appx. A3 reveals an essence of the homomorphism property of the matrix completion problem, The underlying beauty motivates the algorithmic design of Alg. 1 and Alg. 2, which turns out to be quite simple and quite effective.
>
> > Homomorphic encryption is generally supposed to be slow, but the proposed encryption / decryption method using public / private random matrices seems to be quite fast.
>
> Thanks for your positive comment.  Yes, general full homomorphic encryption is quite slow. Our homomorphic version of matrix completion is really fast.
>
>
> > This reviewer found the math in the paper hard to follow and hard to constantly refer to [3] or other references because the paper didn't seem to be self-contained. The tangent cone mentioned after eq (12) is not defined in the paper and neither is the incoherence parameter used in (13).
>
> Sorry that we should clearly point out that the tangent cone $T$ is just the linear space spanned by matrices with the same column space or row space as M. Its definition was put in equation (20) in Appx. A2, which should be moved to right after (12). It was a mistake.
>
> Also, we should move the definition of incoherence parameter, Def. 6 in Appx. A5, to before (13).
>
>
> > Both constants $C$ and $c$ are present in Lemma 3 and Corollary 1. It seems to this reviewer that $C=c$.
>
> In Lemma 3 and Corollary 1, the uppercase $C$ stands for an absolute constant (in ref. [4]), whose value remains the same throughout the proofs. The lowercase $c$ is a normal numerical constant, which may take varying values across different appearances, since the interest is the decaying trend of probability. This confusion is mainly due to the original theoretical papers of matrix completion problems, like ref. [4] and several others.
>
>
> > The parameter $\xi$ in Theorem 4 doesn't seem to be defined anywhere.
>
> Thanks very much! This parameter $\xi$ should be $1- \xi - \delta$, which is an error probability. In the proof of Theorem 4, we employed a two-round encryption, thus we used both $\xi$ and $\delta$. We should have made it clear.
>
>
> > Typo: "Rank-decent" should be "Rank-descent" after eq (12).
>
> Thanks for the careful reading, and the typo is fixed in the revised version.

---

> > ### Author Response · Authors · 2022-08-03
> > **Response to Reviewer WYT8 (2/2)**
> >
> > > The parameter $k$ is a design parameter of the algorithm but the authors simply set it to $10$ in all experiments. Shouldn't it vary with rank $r$ or $p$?
> >
> > The impact of the parameter $k$ can be observed in Fig. 3 directly. The homomorphic matrix completion essentially hides the plaintext $M$ with rank $r$ into a larger space $\overline{M}$ with rank $\overline{r}$, where $\overline{r} \leq r + k$ and $k$ is the dimension of the keys. Therefore, the impact of the parameter $k$ can be observed by varying the rank $r$ shown in Fig. 3, in which a larger $k$ requires more observed entries to achieve exact recovery. An interpretation is also available in Line 293~296.
> >
> > Furthermore, we like to refer to a relevant response for a question raised by ReviewerUWAB:
> >
> > > > One key assumption in matrix completion is low-rankness. However, the transformation M+KR in the server is determined by the public key K to guarantee low rankness. In practice, how do we choose the dimension k since we have no idea the true rank r of M?
> > >
> > > This is an interesting question. It would be quite interesting to point out how a two-round encryption scheme in Appx. B1 would provide a smart way to address the reviewer’s concern.
> > >
> > > * In the first round of encryption, one can choose an arbitrary value  $k’  >= 1$. The server obtains an encrypted $\overline M1$ and estimates a column subspace $K2 = \overline U$ with dimension $k = k’ + r$.
> > > * In the second round of encryption, the server sends both $K2 = \overline U$ and $k= k’+r$ to all user nodes.
> > >
> > > Using the above method, our scheme does not require the true rank $r$ of the data matrix $M$.
> > >
> > > We would like to further discuss real-world applications where the data matrix $M$ is approximately low-rank, i.e., not strictly low-rank with a true rank $r$.  Some works [1, 2, 3] derived a sample complexity with a polynomial dependence on the conditional number $\kappa = \sigma_1 / \sigma_r$.
> > >
> > > If the maximum value of the generated random numbers is larger than $\sigma_1$, or the minimum value is smaller than $\sigma_r$, or both cases hold simultaneously, then those analytical results in [1, 2, 3] need to use a new \kappa value.
> > >
> > > * [1] Jain, Prateek, Praneeth Netrapalli, and Sujay Sanghavi. "Low-rank matrix completion using alternating minimization." Proceedings of the forty-fifth annual ACM symposium on Theory of computing. 2013.
> > > * [2] Hardt, Moritz. "Understanding alternating minimization for matrix completion." 2014 IEEE 55th Annual Symposium on Foundations of Computer Science. IEEE, 2014.
> > > * [3] Keshavan, Raghunandan Hulikal. Efficient algorithms for collaborative filtering. Stanford University, 2012.

---

> > > ### Comment · Reviewer_WYT8 · 2022-08-08
> > > **Response addresses my concerns**
> > >
> > > The authors' response has adequately addressed my concerns and questions.
> > > For some reason, I am not able to see the revised version of the paper or the appendix within OpenReview, but I am quite satisfied with the responses provided.
> > > I wish to thank them for their insightful answer to my question regarding the choice of $k$ via the two-round scheme.
> > > It's also nice to know that the results have been extended to a block-distributed setting in a longer (journal ?) version of this paper.
> > > Even after considering all other reviewer feedback, my positive rating on this paper remains unchanged.

---

### Official Review · Reviewer_UWAB · 2022-07-12

**Rating:** 3
**Confidence:** 4
**Soundness:** 2 fair
**Presentation:** 2 fair
**Contribution:** 2 fair

**Summary:**

This paper studies the problem of privacy-preserving data completion in a distributed manner with the homomorphic matrix completion problem and propose a homomorphic encryption-decryption scheme.

**Questions:**

(1) It is worth discussing whether the holomorphic matrix completion can work in that distributed setting.

(2) One key assumption in matrix completion is low-rankness. However, the transformation M+KR in the server is determined by the public key K to guarantee low rankness. In practice, how do we choose the dimension k since we have no idea the true rank r of M?

(3) The transformation g(M)=M+KR is too standard in DP. Is that possible to apply the non-linear transformation to improve privacy?

(4) Could the authors numerically provide the privacy and recovery trade-off in different privacy levels?

(5) Overall, this work follows a simple idea of differential privacy by replacing random noise as the product of public key and random noise. The guarantee of differential privacy and completion is relatively standard in existing works.

**Limitations:**

Yes

**Strengths And Weaknesses:**

Strengths: The targeted problem is interesting and important. This paper is well organized and easy to follow. Some theoretical results are provided to guarantee recovery and differential privacy.

Weakness:
(1) The centralized server framework in Figure 1(a) does not seem to be a well-known distributed framework in distributed matrix completion. There are too many nodes increasing with the dimension n2. In distributed matrix completion works, say, Mackey, Talwalkar, and Jordan (2015, JMLR), people distribute the matrix into blocks and complete them in parallel. It is worth discussing whether the holomorphic matrix completion can work in that distributed setting.

(2) One key assumption in matrix completion is low-rankness. However, the transformation M+KR in the server is determined by the public key K to guarantee low rankness. In practice, how do we choose the dimension k since we have no idea the true rank r of M?

(3) The transformation g(M)=M+KR is too standard in DP. Is that possible to apply the non-linear transformation to improve privacy?

(4) Could the authors numerically provide the privacy and recovery trade-off in different privacy levels?

(5) Overall, this work follows a simple idea of differential privacy by replacing random noise as the product of public key and random noise. The guarantee of differential privacy and completion is relatively standard in existing works.

---

> ### Author Response · Authors · 2022-08-03
> **Response to Reviewer UWAB (1/2)**
>
> Thank you for your insightful and detailed comments. We would like to address your concerns and your questions in the following.
>
> > It is worth discussing whether the holomorphic matrix completion can work in that distributed setting.
>
> Thanks for suggesting that paper. Yes, our homomorphic framework naturally extends to such a block distributed setting. Actually, we already included that case in our long version (not included in this submission, in order to keep it simple). A block distributed setting, in Fig. 1(a), would be each node (e.g., an edge server in mobile computing) holding multiple columns.
>
> In the following, the authors would like to discuss how the current results have included the “block distributed” scenario as a special case. We will add several sentences (remarks) to clarify it in the revised version.
> * First, for recovery, it does not require any change of Theorem 2/3 and Lemma 3 at all. Consider t nodes and each node has $\ell$ columns, with $n_2 = t * \ell$, as in the paper by Mackey, Talwalkar, and Jordan (2015, JMLR). These variables have the same values: column space $U$, coherence $\mu$ in (13), rank $r$, and $n_2 = t * \ell$.
> * Second, for DP privacy, it may require some change to Theorem 4. In the first scenario where one wants to guarantee user-level privacy, there is no change of Theorem 4. In the second scenario where one wants to guarantee node-level (a node can be an edge server that holds multiple users’ feature vectors) privacy, the sensitivity $\delta$ becomes the maximum Frobenius norm of $n_2$ submatrices, where each submatrix has size $n_1 \times \ell$.
>
>
> > One key assumption in matrix completion is low-rankness. However, the transformation M+KR in the server is determined by the public key K to guarantee low rankness. In practice, how do we choose the dimension k since we have no idea the true rank r of M?
>
> This is an interesting question. It would be quite interesting to point out how a two-round encryption scheme in Appx. B1 would provide a smart way to address the reviewer’s concern.
>
> * In the first round of encryption, one can choose an arbitrary value  $k’  >= 1$. The server obtains an encrypted $\overline M1$ and estimates a column subspace $K2 = \overline U$ with dimension $k = k’ + r$.
> * In the second round of encryption, the server sends both $K2 = \overline U$ and $k= k’+r$ to all user nodes.
>
> Using the above method, our scheme does not require the true rank $r$ of the data matrix $M$.
>
> We would like to further discuss real-world applications where the data matrix $M$ is approximately low-rank, i.e., not strictly low-rank with a true rank $r$.  Some works [1, 2, 3] derived a sample complexity with a polynomial dependence on the conditional number $\kappa = \sigma_1 / \sigma_r$.
>
> If the maximum value of the generated random numbers is larger than $\sigma_1$, or the minimum value is smaller than $\sigma_r$, or both cases hold simultaneously, then those analytical results in [1, 2, 3] need to use a new \kappa value.
>
> * [1] Jain, Prateek, Praneeth Netrapalli, and Sujay Sanghavi. "Low-rank matrix completion using alternating minimization." Proceedings of the forty-fifth annual ACM symposium on Theory of computing. 2013.
> * [2] Hardt, Moritz. "Understanding alternating minimization for matrix completion." 2014 IEEE 55th Annual Symposium on Foundations of Computer Science. IEEE, 2014.
> * [3] Keshavan, Raghunandan Hulikal. Efficient algorithms for collaborative filtering. Stanford University, 2012.
>
>
> > The transformation g(M)=M+KR is too standard in DP. Is that possible to apply the non-linear transformation to improve privacy?
>
> We appreciate your insightful comment. The linear transformation $g(M)=M+KR$ is quite standard in DP.
>
> Consider $g(M)$ where $g$ is a nonlinear function, and $M$’s SVD decomposition $M = USV^T$. Then, we have $g(M) = g(USV^T)$. To be concrete, we can set $g = <f, h>$ where both $f$ and $h$ are linear transforms, i.e.g, $f(U) = AU$ and $h(V) = BV$, thus the result $g$ is bi-linear transforms, since $g(USV^T) = AUSV^TB^T$.
> 1. When $A$ and $B$ are Guassian, their product is “close” to Gaussian [4].
> 2. Projecting $A$ and $B$ into the subspace of $UV^T$ may be equivalent to adding a Gaussian vector to the diagonal matrix $S$.
>
> Therefore, it is possible to apply non-linear transformation and still satisfies the notion of DP privacy. However, the mathematical analysis would be more challenging, which we would like to explore in our long journal version in the future.  Other non-linear transformations besides bi-linear transformations are unclear to us yet.
> * [4] Li, Yi, and David P. Woodruff. "The Product of Gaussian Matrices Is Close to Gaussian." Approximation, Randomization, and Combinatorial Optimization. Algorithms and Techniques (APPROX/RANDOM 2021). Schloss Dagstuhl-Leibniz-Zentrum für Informatik, 2021.

---

> > ### Author Response · Authors · 2022-08-03
> > **Response to Reviewer UWAB (2/2)**
> >
> > > Could the authors numerically provide the privacy and recovery trade-off in different privacy levels?
> >
> > This is a challenging question! The primary finding is that low-rank matrix does not exhibit a privacy-recovery tradeoff, but a privacy-sample tradeoff, because the proposed homomorphic framework can achieve EXACT recovery at a cost of more samples.
> >
> > However, numerical experiments may still help investigate the privacy-recovery tradeoff at different privacy levels:
> > 1. Given different privacy levels, according to Theorem 4, the epsilon-delta parameters will determine the standard deviation of the Guassian distribution.
> > 2. The added Guassian random numbers will affect the recovery via the condition number. As described in the above that if the maximum value of the generated random numbers is larger than $\sigma_1$, or the minimum value is smaller than $\sigma_r$, or both cases hold simultaneously, then those analytical results in [1, 2, 3] need to use a new $\kappa$ value.
> >
> > We are considering adding some numerical results in an appendix, in order to give readers a better understanding of whether the privacy-recovery trade-off has fully vanished.
> >
> > > Overall, this work follows a simple idea of differential privacy by replacing random noise as the product of public key and random noise. The guarantee of differential privacy and completion is relatively standard in existing works.
> >
> > Your summary is accurate.
> >
> > Moreover, the authors would like to say that Alg. 1~2 presents an add-on scheme to existing matrix completion methods, which is relatively easy to implement. Quite unexpected is that it guarantees both EXACT recovery and DP property. Therefore, we would like to share this finding to the community.

---

> > > ### Author Response · Authors · 2022-08-09
> > > **Any feedback to the responses from the authors?**
> > >
> > > The reviewer (Reviewer UWAB) raised an interesting question about the “block distributed” setting, which is a special case covered by the proposed homomorphic encryption framework, and the provided theoretical results naturally apply.
> > >
> > > About ​​parameter k and the true rank r of data matrix M, the authors restate the two-round scheme (given in the Appx. B1) in the responses, which does not require prior knowledge of rank r or proper selection of parameter k.
> > >
> > > Would like to know whether Reviewer UWAB thinks that the EXACT recovery (the novel homomorphism property) of matrix completion is straightforward.
> > >
> > > Thanks very much!

---

### Official Review · Reviewer_ux8W · 2022-07-30

**Rating:** 6
**Confidence:** 2
**Soundness:** 3 good
**Presentation:** 2 fair
**Contribution:** 2 fair

**Summary:**

This paper presents a private and secure matrix completion scheme, where the data gets sent to a server for the task. The communication security is based on homomorphic encryption, and the privacy notion here is $(\varepsilon,\delta)$-differential privacy (DP). The authors work with a relaxed notion of joint DP.

Both theoretical and experimental results are provided for completeness. The focus is on exact recovery in this paper.

Edit: Score updated.

**Questions:**

I don't necessarily have many questions. Could I know what the assumptions behind the data are? I would like to see what the sensitivity for the added noise is. How big is the norm bound $L$ in practice, and how is it affecting the accuracy of your scheme?

What is the running time of the scheme compared to prior work?

**Limitations:**

I couldn't find where the authors discuss the limitations of their work. Could I be pointed to that, please?

I feel that the writing quality could be improved by adding more detailed preliminaries for the main problem and the cryptographic primitives (even in the supplement).

In Equation 13, what is $\mu(U)$?

There seems to be some typo at the end of Line 130.

Lines 114-115: What does $\mathcal{A}-j(D)$ mean? How is this different from running $\mathcal{A}$ on $D_{-j}$?

Lines 102-106: there are both bullets and numbering.

**Strengths And Weaknesses:**

Strengths:
The paper does provide a scheme that seems to satisfy both the communication security and DP requirements. This paper actually focuses on exact recovery, as opposed to sacrificing accuracy in the prior work. The experimental results indicate compatibility with two matrix completion algorithms, NN and AM.

Weaknesses:
1. I wish the writing could be a bit more clear in terms of the cryptographic tools and the preliminaries related to matrix completion problem.
2. The DP side of the scheme doesn't seem very interesting as it just involves adding some Gaussian noise. I don't see any assumptions about the data range (except that the $\ell_2$ norm is bounded by $L$), or how they're being removed. I would have liked to see more of that. This problem might have just required something that simplistic, but I certainly wouldn't highlight its DP property as a huge selling-point for this reason.

---

> ### Author Response · Authors · 2022-08-03
> **Response to Reviewer ux8W (1/2)**
>
> Thank you for your thoughtful comments. We would like to address your concerns and your questions in the following.
>
> > I wish the writing could be a bit more clear in terms of the cryptographic tools and the preliminaries related to matrix completion problem.
>
> Appreciate your feedback. We plan to add a background of cryptographic tools and the matrix completion problem in both problem setting and appendix, to make the paper easy to follow.
>
> > The DP side of the scheme doesn't seem very interesting as it just involves adding some Gaussian noise. I don't see any assumptions about the data range (except that the $\ell_2$ norm is bounded by $L$), or how they're being removed. I would have liked to see more of that. This problem might have just required something that simplistic, but I certainly wouldn't highlight its DP property as a huge selling-point for this reason.
>
> We agree with your evaluation. Regarding DP property, there are two points:
> 1. We propose a variant of DP notation for low-rank matrices, since we believe it is necessary to exclude the shared column subspace, which is treated as common information;
> 2. Yes, the DP side of Alg. 1~2  is simply the standard Gaussian mechanism (additive Gaussian noise), and it is not a selling-point.
>
> Moreover, the authors would like to say that Alg. 1~2 presents an add-on scheme to existing matrix completion methods, which is relatively easy to implement. Quite unexpected is that it guarantees both EXACT recovery and DP property. Therefore, we would like to share this finding to the community.
>
>
> > Could I know what the assumptions behind the data are?
>
>  Thanks for your questions. They are important in practical usage! Our responses are drawn from two aspects:
> 1. Algorithmic requirement to guarantee data recovery;
> 1. Algorithmic requirement to achieve a relatively high level of data privacy.
>
> Regarding data assumption, the underlying assumptions are
> 1. The actual data matrix $M$ is low-rank, which means that truncating the SVD decomposition of $M$ using a small rank (say $r$ is one order smaller than $n$, if $n=10000$, then $r = 1000$) can get a good estimate.
> 1. A relatively small coherence value $\mu$, which means that a user’s preference ratings are not concentrated on a few entries but spread out across relatively many entries.
>
>
> > I would like to see what the sensitivity for the added noise is.
>
> The sensitivity for the matrix completion problem is the norm bound $L = ||M_j ||$ (the $L2$ norm of a matrix column), since the algorithm is expected to output the true data matrix; then the sensitivity (under DP notion) of the algorithm is, $\delta = argmax ||A(D) - A(D’)||_2 = L$, since the $L2$ difference between $A(D)$ and $A(D’)$ is upper bounded by the maximum norm of $M$’s columns.
>
> > How big is the norm bound $L$ in practice, and how is it affecting the accuracy of your scheme?
>
> We would like to elaborate more about $L$’s value in practical scenarios and how it affects the scheme performance.  We checked the following datasets in ref. [14, 28].
>
> Recommendation systems:
> 1. MovieLens (Top 400) [14], or $10^5$ , $10^6$ and $10^7$ in [28];
> 1. Netflix (Top 400) in [14];
> 1. Jester;
> 1. Yahoo! Music.
>
> In most recommendation systems, the rating matrix takes values in {1, 2, 3, 4, 5}. So, in practice, the norm bound $L$ depends on the size of rows (user’s feature vector), say $5 \sqrt{n_1}$, which is $500$ for $n_1 = 10,000$.
>
> Furthermore, $L$ together with $\epsilon$ and $\delta$ will directly determine the standard deviation of the Gaussian distribution, and thus may affect the accuracy of the proposed scheme through a condition number of the encrypted matrix. It would be good to include more numerical results in the appendix, in order to provide readers a sense of how $L$ affects the algorithms’ performance. The relationship is as follows:
>
> 1. Some works [1, 2, 3] derived a sample complexity with a polynomial dependence on the conditional number $\kappa = \sigma_1 / \sigma_r$.
> 1. If the maximum value of the generated random numbers is larger than $\sigma_1$, or the minimum value is smaller than $\sigma_r$, or both cases hold simultaneously, then those analytical results in [1, 2, 3] need to use a new $\kappa$ value.
>
> * [1] Jain, Prateek, Praneeth Netrapalli, and Sujay Sanghavi. "Low-rank matrix completion using alternating minimization." Proceedings of the forty-fifth annual ACM symposium on Theory of computing. 2013.
> * [2] Hardt, Moritz. "Understanding alternating minimization for matrix completion." 2014 IEEE 55th Annual Symposium on Foundations of Computer Science. IEEE, 2014.
> * [3] Keshavan, Raghunandan Hulikal. Efficient algorithms for collaborative filtering. Stanford University, 2012.

---

> > ### Author Response · Authors · 2022-08-03
> > **Response to Reviewer ux8W (2/2)**
> >
> > > What is the running time of the scheme compared to prior work?
> >
> > We appreciate your question and would be happy to clarify it. The encryption method can be simply understood as $g(M)=M+KR$, whose computation involves the following four steps:
> > 1. A server generates a matrix $K$;
> > 1. Each node locally generates a random vector and performs one matrix multiplication $KR$ and one matrix addition $M + KR$;
> > 1. A server performs a matrix completion on $M + KR$ with missing entries indicated by $\Omega$;
> > 1. Each node locally performs a matrix subtraction $M - KR$.
> >
> > The above step (3) uses conventional matrix completion methods, which may take a long running time for large matrices.  Our method introduces Step (1),  (2) and (4), whose computations are relatively simple. Also, in our experiments, our method almost does not introduce extra running time.
> >
> >
> > > I couldn't find where the authors discuss the limitations of their work. Could I be pointed to that, please?
> >
> > We think the following positions can be some potential limitations of this work:
> > 1. Please compare Def. 3 with Def. 2 and explanations in Lines 124~130.  Here, we propose to relax the conventional privacy notation to a subspace-aware counterpart. We hold the assumption that the shared column subspace is common information that may not need to be protected. Also, this is possible since we are targeting a distributed matrix completion problem.
> > 2. Similarly, the target problem of distributed matrix completion may not fit many real-world recommendation systems. For example, for current recommendation systems that collect data from all users and then perform data recovery; here, our framework proposes that future recommendation systems should take a distributed structure, where a user’s local app has her own data vector and only sends encrypted data to a central server. Such a shift is not well-supported by existing systems yet.
> >
> >
> > > I feel that the writing quality could be improved by adding more detailed preliminaries for the main problem and the cryptographic primitives (even in the supplement).
> >
> > We appreciate your feedback very much and will polish the writing in the revised version.
> >
> > > In Equation 13, what is $\mu(U)$?
> >
> > $\mu(U)$ is the coherence measure of $U$, where $U$ is the $r$ left singular vector of $M$. We have added the definition of the coherence measure in the revised version (Line 201~204).
> >
> > > There seems to be some typo at the end of Line 130.
> >
> > Thanks for the careful reading, and the typo is fixed in the revised version.
> >
> > > Lines 114-115: What does $A-j(D)$ mean? How is this different from running $A$ on $D_{-j}$?
> >
> > $A−j(D)$ means that excluding the $j$-th output of algorithm $A$, while running $A$ on $D−j$ means excluding the $j$-th input of algorithm $A$.
> >
> > > Lines 102-106: there are both bullets and numbering.
> >
> > The format is corrected in the revised version.

---

### Author Response · Authors · 2022-08-08
**Summary and Thanks to All Reviewers and Area Chair**

The authors sincerely thank all reviewers and area chair. To recap, this work has made the following major contributions.

1. A novel homomorphic encryption framework for the distributed matrix completion problem. In contrast to conventional homomorphic encryption methods that are slow, the proposed scheme is quite fast.
2. With a similar level of $(\epsilon, \delta)$ differential privacy, this paper improves the existing best-known error bound $O(\sqrt[10]{n_1^3n_2})$ in [14] to EXACT recovery at a cost of more samples. It changes the conventional utility (accuracy)-privacy tradeoff to a novel #samples-privacy tradeoff.
3. The reviewer (Reviewer UWAB) raised an interesting question about the “block distributed” setting, which is a special case covered by the proposed homomorphic encryption framework, and the provided theoretical results naturally apply.
4. Both Reviewer WYT8 and Reviewer UWAB raise questions about ​​parameter $k$ and the true rank $r$ of data matrix $M$. The authors restate the two-round scheme (given in the Appx. B1) in the responses, which does not require prior knowledge of rank $r$ or proper selection of parameter $k$. It will be made clear in the paper.

Reviewer UWAB is right about the underlying idea of Alg. 1 and Alg. 2 that replaces random noise of differential privacy as the product of public key and random noise. However, the EXACT recovery (the novel homomorphism property) of matrix completion is non-trivial.

---

### Meta-Review · Area_Chair_P8pm · 2022-08-22

**Recommendation:** Accept
**Confidence:** Less certain

**Metareview:**


This paper concerns privacy-preserving matrix completion in a distributed manner. The communication security is based on homomorphic encryption, while the notion of privacy is defined as the subspace-aware join differential privacy.

The paper received a mixed evaluation from the reviewers, ranging from accept (7) to reject (3), and the reviewers that gave these scores decided to keep them after the rebuttal and the following discussion.

The strengths of the paper mentioned by the reviewers were:
- Focusing on an interesting and important problem
- Providing an algorithm that guarantees the exact recovery, as opposed to sacrificing accuracy in the prior work
- Solid experimental results

On the other hand, the identified weaknesses were:
- The DP side does not seem very interesting, just involving Gaussian mechanism
- Necessity to know at least an estimate of the rank of M before the homomorphic encryption (a 2-phase solution to that is sketched in the rebuttal)
- The paper not being self-contained
- Some technical issues, which (I believe) were clarified in the feedback

Despite the weaknesses mentioned above, I lean toward the acceptance with my recommendation, although with a limited confidence.

**Award:**

No

---

### Decision · Program_Chairs · 2022-09-14

Accept